# Cocaine-induced endocannabinoid signaling mediated by sigma-1 receptors and extracellular vesicle secretion

Yoki Nakamura[1†‡], Dilyan I Dryanovski[2†], Yuriko Kimura[1], Shelley N Jackson[3], Amina S Woods[3], Yuko Yasui[1], Shang-Yi Tsai[1], Sachin Patel[1,4], Daniel P Covey[5], Tsung-Ping Su[1]*, Carl R Lupica[2]*

[1]Cellular Pathobiology Section, Intramural Research Program, National Institute on Drug Abuse, National Institutes of Health, Baltimore, United States; [2]Electrophysiology Research Section, Intramural Research Program, National Institute on Drug Abuse, National Institutes of Health, Baltimore, United States; [3]Structural Biology Unit, Intramural Research Program, National Institute on Drug Abuse, National Institutes of Health, Baltimore, United States; [4]Department of Psychiatry and Behavioral Sciences, Vanderbilt Brain Institute, Vanderbilt University Medical Center, Vanderbilt University, Nashville, United States; [5]Department of Anatomy and Neurobiology, University of Maryland School of Medicine, Baltimore, United States

*For correspondence:
TSU@intra.nida.nih.gov (T-PS);
clupica@mail.nih.gov (CRL)

[†]These authors contributed equally to this work

Present address: [‡]Graduate School of Biomedical & Health Sciences, Hiroshima University, Hiroshima, Japan

Competing interests: The authors declare that no competing interests exist.

**Abstract** Cocaine is an addictive drug that acts in brain reward areas. Recent evidence suggests that cocaine stimulates synthesis of the endocannabinoid 2-arachidonoylglycerol (2-AG) in midbrain, increasing dopamine neuron activity via disinhibition. Although a mechanism for cocaine-stimulated 2-AG synthesis is known, our understanding of 2-AG release is limited. In NG108 cells and mouse midbrain tissue, we find that 2-AG is localized in non-synaptic extracellular vesicles (EVs) that are secreted in the presence of cocaine via interaction with the chaperone protein sigma-1 receptor (Sig-1R). The release of EVs occurs when cocaine causes dissociation of the Sig-1R from ADP-ribosylation factor (ARF6), a G-protein regulating EV trafficking, leading to activation of myosin light chain kinase (MLCK). Blockade of Sig-1R function, or inhibition of ARF6 or MLCK also prevented cocaine-induced EV release and cocaine-stimulated 2-AG-modulation of inhibitory synapses in DA neurons. Our results implicate the Sig-1R-ARF6 complex in control of EV release and demonstrate that cocaine-mediated 2-AG release can occur via EVs.
DOI: https://doi.org/10.7554/eLife.47209.001

## Introduction

The sigma-1 receptor (Sig-1R) is a small protein that resides at the endoplasmic reticulum (ER)-mitochondrion interface (mitochondrion-associated ER membrane; MAM) (**Hayashi and Su, 2007**; **Hayashi et al., 2009**; **Mori et al., 2013**), where it constrains type-3 inositol 1,4,5-trisphosphate receptors (IP$_3$R3) to facilitate Ca$^{2+}$ signaling from ER to mitochondria (**Hayashi and Su, 2007**; **Hayashi et al., 2009**). In addition, the Sig-1R binds a wide range of molecules, including psychotropic drugs and psychostimulants, such as cocaine and methamphetamine (**Largent et al., 1987**), and can translocate to other cellular regions to associate with organelles, proteins, plasma membranes, and the nuclear envelope to control trafficking of other molecules, such as ion channels and receptors in neurons (**Su et al., 2016**; **Yasui and Su, 2016**). These diverse signaling roles for the Sig-1R highlight a widespread influence on cellular function that is incompletely understood.

**eLife digest** The cannabis plant contains hundreds of different chemicals, including more than sixty types of cannabinoids. By binding to specific sites on brain cells, cannabinoids change how cells communicate with one another. This in turn triggers widespread alterations in brain activity, which can affect mood, appetite, coordination and perception.

But not all cannabinoids come from plants. The brain also produces its own versions, known as endocannabinoids (or eCBs for short). These bind to the same sites on brain cells as the plant-derived chemicals. Changes in endocannabinoid activity have been implicated in various brain disorders. These include Alzheimer's disease, epilepsy and stress disorders. They may also have a role in drug addiction. Exposing rats to cocaine causes endocannabinoid levels to increase in areas of the brain that process pleasurable sensations. This suggests that the release of endocannabinoids may contribute to cocaine addiction. But how cocaine triggers this release has been unclear.

By studying brain tissues and cells kept alive in petri dishes, Nakamura, Dryanovski et al. show that cocaine drives cells to release endocannabinoids via a process called extracellular vesicle release. In essence, cocaine causes cells to make endocannabinoids that are then enclosed inside membrane-bound packages. These packages – or extracellular vesicles – can then fuse with the cell's outer membrane. Multiple proteins must interact with each other for cells to assemble and release extracellular vesicles. Nakamura, Dryanovski et al. show that disrupting these interactions prevents vesicles from forming, and also prevents cocaine from triggering endocannabinoid release. Blocking extracellular vesicle release prevents cocaine from altering communication between brain cells.

Cocaine thus drives endocannabinoid release in the brain's pleasure centers via the assembly of extracellular vesicles. Using other drugs to manipulate the protein interactions that underlie vesicle assembly could provide a new way to counter cocaine addiction.

DOI: https://doi.org/10.7554/eLife.47209.002

Substantial data suggest that the Sig-1R is also a target of the abused psychostimulant cocaine (*Hayashi and Su, 2007*; *Sharkey et al., 1988*; *Hayashi and Su, 2003*; *Kourrich et al., 2013*; *Tsai et al., 2015*; *Chen et al., 2007*). In the mouse nucleus accumbens (NAc), cocaine decreases the excitability of GABAergic medium spiny neurons by strengthening an association between the Sig-1R and Kv1.2 potassium channels, contributing to behavioral sensitization to the drug (*Kourrich et al., 2013*). Moreover, the Sig-1R is also involved in cocaine reward (*Romieu et al., 2002*). Given the diverse demonstrated roles for the Sig-1R in cellular signaling, its regulation by cocaine has the potential to affect many unknown cellular properties.

Extracellular vesicles (EVs) are a diverse group of membranous entities of endosomal origin that are secreted from a broad range of cell types (*van Niel et al., 2018*). The EV classification broadly includes exosomes and microvesicles that range in size from 30 to 150 nm, and 100–1000 nm, respectively. Exosomes are formed by invagination of the endosomal membrane to form multivesicular bodies that are released into the extracellular space via budding of the cellular membrane, whereas microvesicles are formed by budding of plasma membrane (*van Niel et al., 2018*; *Radhakrishna et al., 1996*; *Huang-Doran et al., 2017*). It is increasingly apparent that EV formation occurs through highly regulated cellular processes (*Abels and Breakefield, 2016*), that permit their participation in intercellular communication via delivery of cargos of RNAs, microRNAs, proteins, and bioactive lipids such as prostaglandins (*van Niel et al., 2018*; *Huang-Doran et al., 2017*; *EL Andaloussi et al., 2013*). This implicates EVs in a wide range of physiological and pathological processes. EV motility can be controlled by signaling molecules such as the guanine-nucleotide binding protein, ADP-ribosylation factor 6 (ARF6) (*Abels and Breakefield, 2016*; *Muralidharan-Chari et al., 2009*; *D'Souza-Schorey and Chavrier, 2006*). As a small GTPase, ARF6 exists in GTP- or GDP-bound forms (ARF6-GTP or ARF6-GDP), and stimulation of ARF6 by neurotransmitters or growth factors recruits guanine nucleotide exchange factors (GEFs) to convert ARF6-GDP to the active ARF6-GTP (*EL Andaloussi et al., 2013*). Although ARF6 itself has GTPase activity, ARF6-GTP requires GTPase-activating proteins (GAPs) to hydrolyze to its inactive ARF6-GDP form. ARF6-GTP influences a wide variety of cellular events including endocytosis, actin cytoskeleton reorganization

and phosphoinositide metabolism in many types of cells. Importantly, ARF6-GTP is involved in EV release from plasma membranes (*Muralidharan-Chari et al., 2009*; *Than et al., 2017*), and exosome budding into multivesicular bodies (*Ghossoub et al., 2014*; *Friand et al., 2015*; *Imjeti et al., 2017*). Thus, GEFs and GAPs regulate ARF6 activity to then modulate EV secretion (*D'Souza-Schorey and Chavrier, 2006*). Also, since ARF6 is a GTPase, it is noteworthy that another GTPase, Rac-GTPase, forms a complex with the Sig-1R (*Natsvlishvili et al., 2015*), suggesting the possibility that the Sig-1R may interact with other molecules of this class. Collectively, these points of regulation position EVs and ARF6 as important participants in diverse physiological and pathological processes (*van Niel et al., 2018*).

Endocannabinoids (eCB) are lipid signaling molecules that activate CB1 or CB2 cannabinoid receptors. Of these, CB1Rs are expressed at high levels on neuronal axon terminals where they inhibit fast neurotransmitter release (*Misner and Sullivan, 1999*; *Hoffman and Lupica, 2000*; *Katona et al., 1999*). The eCBs are typically synthesized in postsynaptic structures, such as dendrites, to then retrogradely activate CB1Rs on axon terminals (*Wilson and Nicoll, 2001*). Moreover, eCBs are not released via canonical mechanisms of calcium-dependent synaptic vesicle exocytosis, but rather through poorly understood processes. Recent evidence gathered using cell cultures suggests that the eCB *N*-arachidonoylethanolamine (AEA, anandamide) is found in EVs, suggesting a possible mechanism to release these messengers and permit retrograde eCB signaling (*Gabrielli et al., 2015a*; *Gabrielli et al., 2015b*). Another eCB, 2-arachidonoylglycerol (2-AG), is released from neurons in an activity-dependent fashion, or via neurotransmitter stimulation of phospholipase-regulating G-protein coupled receptors (GPCRs) (*Kano et al., 2009*; *Maejima et al., 2005*; *Alger and Kim, 2011*). Recent evidence shows that inhibition of catecholamine uptake by cocaine leads to activation of GPCRs that stimulate 2-AG synthesis in the rodent ventral tegmental area (VTA) (*Wang et al., 2015*). Moreover, as VTA GABAergic axons express CB1Rs, the cocaine-stimulated increase in 2-AG inhibits GABA release via these receptors (*Wang et al., 2015*; *Riegel and Lupica, 2004*), and this can be used as a sensitive measure of eCB function. Although measurements like these are used to detect eCBs throughout the CNS, the mechanisms through which these lipids cross the extracellular space to bind to presynaptic CB1Rs remain poorly understood.

Given that cocaine stimulates 2-AG synthesis, can act as a Sig-1R agonist, and that the Sig-1R interacts with Rac-GTPase, we hypothesize that it may also control other GTPases such as ARF6, a known EV release modulator (*Muralidharan-Chari et al., 2009*; *Ghossoub et al., 2014*; *Natsvlishvili et al., 2015*; *Tsai et al., 2009*), and this might regulate 2-AG release. Through convergent experiments we demonstrate that Sig-1Rs can control EV release via interaction with ARF-6, and that cocaine stimulates this process. Moreover, the cocaine-evoked 2-AG release required intact Sig-1Rs, ARF-6, and cytoskeletal function, implicating EVs as a mechanism for 2-AG release in the VTA.

## Results

### Cocaine activation of Sig-1Rs stimulates EV release from NG-108 cells

To investigate whether Sig-1Rs are involved in EV function, we first conducted studies in NG-108 cells to permit manipulation of signaling pathways. The integrin β1 (Iβ1; CD29) protein mediates transcellular interaction of EVs with target membranes, and is a useful marker of EVs isolated through differential sequential sucrose-gradient centrifugation (*van Niel et al., 2018*; *EL Andaloussi et al., 2013*; *Muralidharan-Chari et al., 2009*; *Imjeti et al., 2017*; *Benmoussa et al., 2019*; *Momen-Heravi et al., 2013*). We prepared membrane fractions enriched in EVs in effluent from NG-108 cells and measured Iβ1 using western blots. Cocaine (10 μM) caused a time- and concentration-dependent increase in the accumulation of Iβ1 in isolated fractions from these NG-108 cells (*Figure 1A and B*), suggesting that cocaine increased EV release. Because our previous studies show that cocaine interacts with Sig-1Rs, we next investigated their involvement in cocaine-stimulated EV release. We found that the Sig-1R agonists PRE-084, or fluvoxamine, both increased the Iβ1-marker of EV release from NG-108 cells in the absence of cocaine (*Figure 1B*), and that pretreatment with either of the Sig-1R antagonists, BD1063 (*Figure 1C*) or NE100 (*Figure 1D*), prevented the effect of cocaine. We also found that the knock-down of Sig-1Rs with siRNA alone significantly increased Iβ1

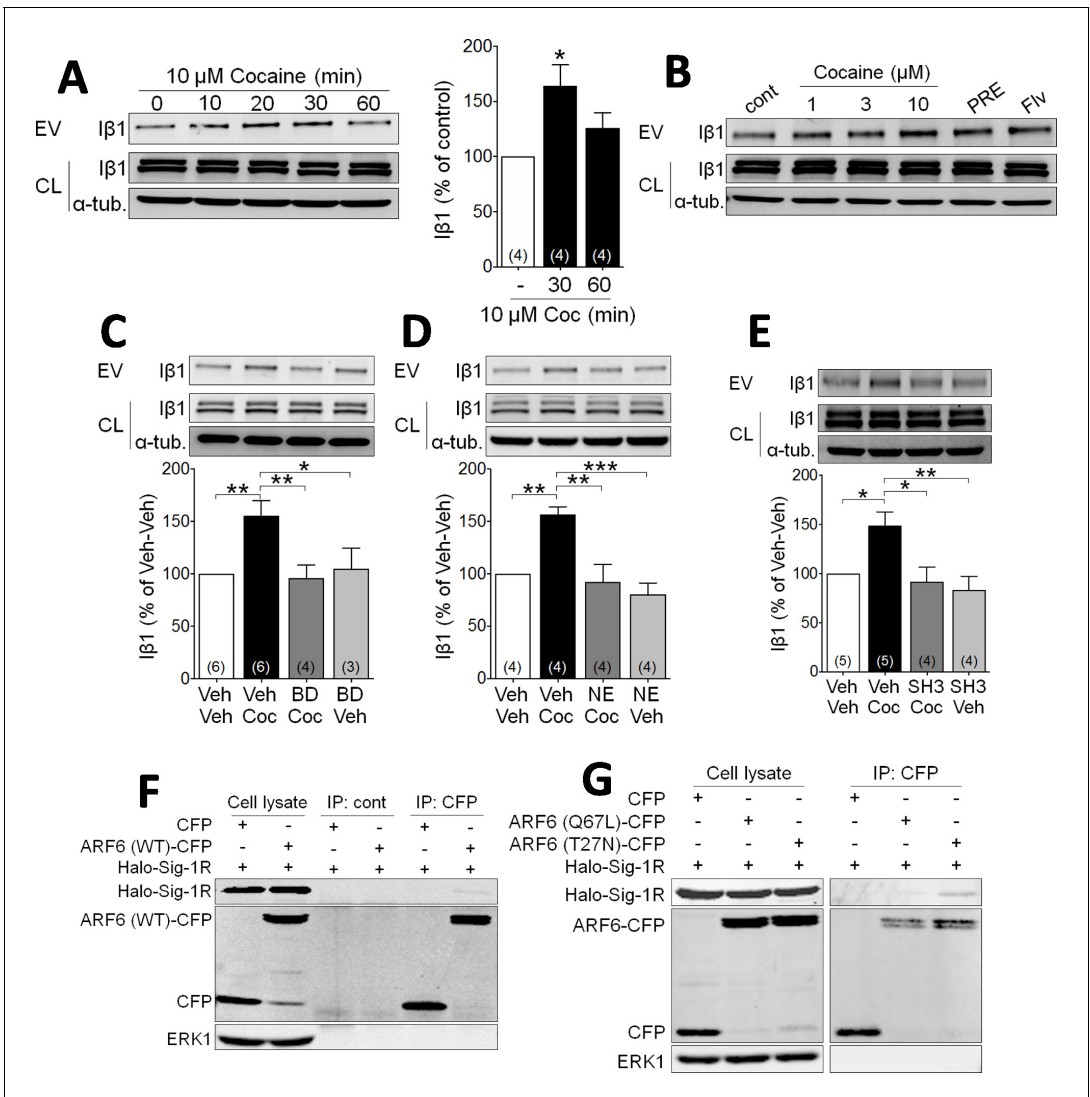

**Figure 1.** Cocaine stimulates EV release via Sig-1R and ARF6 signaling in NG108 Cells. (**A**) Effect of cocaine (10 μM) on integrin β1 (Iβ1) concentration in EV-rich fractions of NG-108 cells at several time points. Western blots also show relative amounts of Iβ1 in cell lysates (CL), and α-tubulin protein (α-tub.) as a control. Bar graph showing the relative change in Iβ1 at 30 and 60 min after cocaine treatment (mean ± S.E.M, $F_{2,9}$ = 5.7, p=0.026, one-way ANOVA, *=p < 0.05 compared to control, Dunnett's multiple comparison test). The number of replications of the experiment at left is shown in parentheses for each group in the bar graph. (**B**) Concentration-dependent effect of cocaine, and effects of the Sig-1R agonists, PRE084 (PRE, 1 μM), and Fluvoxamine (Flv, 10 μM) on Iβ1 concentration in the NG-108 cell culture media, 30 min after treatments (n = 1). (**C–D**) Sig-1R antagonists prevent cocaine-induced EV release in NG-108 cells. BD1063 (BD, 1 μM) or NE100 (NE, 1 μM) were applied to NG-108 cell cultures 10 min before cocaine treatment (C: means ± S.E.M, $F_{3,15}$ = 6.2, p=0.006, one-way ANOVA, *=p < 0.05, **=p < 0.01, Dunnett's multiple comparison test; D: means ± S.E.M, $F_{3,12}$ = 10.4, p=0.001, one-way ANOVA, *=p < 0.05, **=p < 0.01, ***=p < 0.001, Dunnett's multiple comparison test). (**E**) Inhibition of ARF-6 activation by the GEF inhibitor, SH3 (10 μM) blocks the increase in EV release caused by cocaine in NG-108 cells. Cocaine (10 μM) was applied for 30 min, beginning 10 min after SH3 application (n = 4; means ± S.E.M, $F_{3,14}$ = 6.5, p=0.005, one-way ANOVA, *=p < 0.05, **=p < 0.01, Dunnett's multiple comparison test). (**F**) Immunoprecipitation of the Sig-1R/ARF6 complex. Halo-Sig-1R was co-transfected with either cyan-fluorescent protein (CFP) and ARF6 (WT)-CFP into NG108 cells (n = 1). (**G**) Sig-1R prefers ARF6 inactive form. Halo-Sig-1R was co-transfected with CFP, ARF6 (Q67L: mimicking ARF6-GTP)-CFP, or ARF6 (T27N: mimicking ARF6-GDP)-CFP into NG108 cells and co-immunoprecipitation studies performed (n = 1). The number of replications of each experiment is shown in parentheses for each group in the bar graphs. See *Source data 1* for values used in statistical analyses. *Figure 1—figure supplement 1* shows that Sig-1R knockdown alters cocaine effects on EV release as well as the identification of the ARF6 binding site in NG108 cells.

DOI: https://doi.org/10.7554/eLife.47209.003

The following figure supplement is available for figure 1:

**Figure supplement 1.** Sig-1R knockdown alters cocaine effects on EV release and identification of the ARF6 binding site in NG108 cells.

DOI: https://doi.org/10.7554/eLife.47209.004

and abolished the stimulatory effect of cocaine (*Figure 1—figure supplement 1A*), and that overexpression of Halo-tagged Sig-1Rs decreased EV release from NG-108 cells, but also blocked the effect of cocaine (*Figure 1—figure supplement 1B*). These data support a mechanism in which Sig-1Rs tonically inhibit EV release, and this inhibition is relieved in the presence of cocaine. Having established that Sig-1Rs are involved in the stimulatory effect of cocaine on EV release in NG-108 cells, we next investigated the role of additional other signaling molecules known to also regulate EV secretion (*van Niel et al., 2018*; *Muralidharan-Chari et al., 2009*; *Imjeti et al., 2017*).

Cytohesins are a family of GEFs that activate ARFs by catalyzing a shift from GDP- to GTP-bound forms (*D'Souza-Schorey and Chavrier, 2006*; *Frank et al., 1998*; *Hafner et al., 2006*), and this can trigger EV release from LOX cells (*D'Souza-Schorey and Chavrier, 2006*; *Than et al., 2017*). To determine whether ARF6 is similarly involved in cocaine-induced release of EVs in NG108 cells, we used the GEF inhibitor secinH3 (SH3, 10 μM) (*Hafner et al., 2006*) and found that it prevented the cocaine-stimulated increase in Iβ1 levels in the EV fractions (*Figure 1E*). To next determine the nature of the association between ARF6 and Sig-1R proteins in NG-108 cells, we overexpressed ARF6 mutants that mimic either the active, GTP-bound (Q67L), or the inactive GDP-bound (T27N) forms of this protein, and performed co-immunoprecipitation experiments with a Halo-tagged Sig-1R (Halo-Sig-1R) (*Radhakrishna et al., 1996*; *Muralidharan-Chari et al., 2009*). We found that the Halo-Sig-1R co-immunoprecipitated much more strongly with the GDP-bound form of ARF6 (ARF6-T27N), compared to either wild-type ARF6, or the GTP-bound form (ARF6-Q67L) (*Figure 1F*, *Figure 1G*). This suggests that the Sig-1R more strongly binds the inactive GDP-ARF6, rather than the active GTP-ARF6.

As previous studies show that the Sig-1R C-terminus region contains a chaperone domain that interacts with MAM proteins (*Hayashi and Su, 2007*; *Su et al., 2016*; *Ortega-Roldan et al., 2013*), we also performed experiments with mutant Sig-1Rs to determine the regions of interaction with ARF6-GDP (*Figure 1—figure supplement 1C*). NG-108 cells were transfected with plasmids expressing Halo-tagged N- or C-termini on the full-length Sig-1R (Halo-Sig-1R and Sig-1R-Halo, respectively), or on truncated forms of the Sig-1R (Sig-1R-1–60-Halo or Halo-Sig-1R-61–223) that contained chaperone (*Hayashi and Su, 2007*), or ligand binding motifs (*Chen et al., 2007*; *Pal et al., 2008*), respectively. We then examined whether the Halo-tagged receptors co-immunoprecipitated with either the active or the inactive ARF6 mutants described above. The inactive form of ARF6 (ARF6-T27N) co-precipitated with Sig-1R-61–223-Halo, but not with Sig-1R-1–60-Halo (*Figure 1—figure supplement 1C*), suggesting that the C-terminus, chaperone region of the Sig-1R interacts with GDP-bound ARF6. Interestingly, co-immunoprecipitation also revealed that ARF6-T27N interacted with the Halo-Sig-1R, but not the Sig-1R-Halo (*Figure 1—figure supplement 1C*), suggesting that the C-terminus tag interferes with the interaction between Sig-1R and ARF6.

Taken together, our data in NG-108 cells support a model in which the chaperone region of the Sig-1R binds to the inactive form of ARF6 (GDP-ARF6) to tonically inhibit EV release. Therefore, we next examined the co-localization of ARF6 and Sig-1Rs and their ability to regulate EV release in the mouse midbrain to determine the functional relevance of this interaction.

## Sig-1Rs mediate effects of cocaine on EV release in mouse midbrain

Mice received single injections of cocaine (15 mg/kg, i.p.), followed by removal and processing of the midbrain for EV content (*Figure 2—figure supplement 1*). In agreement with previous reports (*Perez-Gonzalez et al., 2012*; *Polanco et al., 2016*), a membrane fraction 3 (fr3), obtained by sequential sucrose-gradient centrifugation, was isolated and found to be enriched with several markers of EVs, such as Iβ1, alix, and flotillin-1 (*Figure 2A*). Moreover, high concentrations of ARF6 and tyrosine hydroxylase (TH) were found in the EV enriched fr3 (*Figure 2A*). However, because of the stringency of the EV isolation procedure, only a small amount of material could be obtained for analysis from these fractions. Therefore, in several experiments, we also utilized a total EV membrane fraction preparation (tEV) that was not subjected to a stepwise sucrose gradient, but nevertheless contained the same EV markers as fr3 (*Figure 2—figure supplement 1*). The mean size of the midbrain tEVs was 154 ± 1.41 nm (*Figure 2C*), and midbrain tEVs contained higher levels of Iβ1, ARF6, and TH, compared to tEVs isolated from cortex and hippocampus (*Figure 2B*).

The topology of TH, Iβ1, and ARF6 in midbrain tEV preparations was next examined using the broad-spectrum serine protease, proteinase-K (PK) (*Wang et al., 2017*; *de Jong et al., 2016*). In tEVs not treated with Triton X detergent, PK decreased only Iβ1 (*Figure 2D*), which is consistent

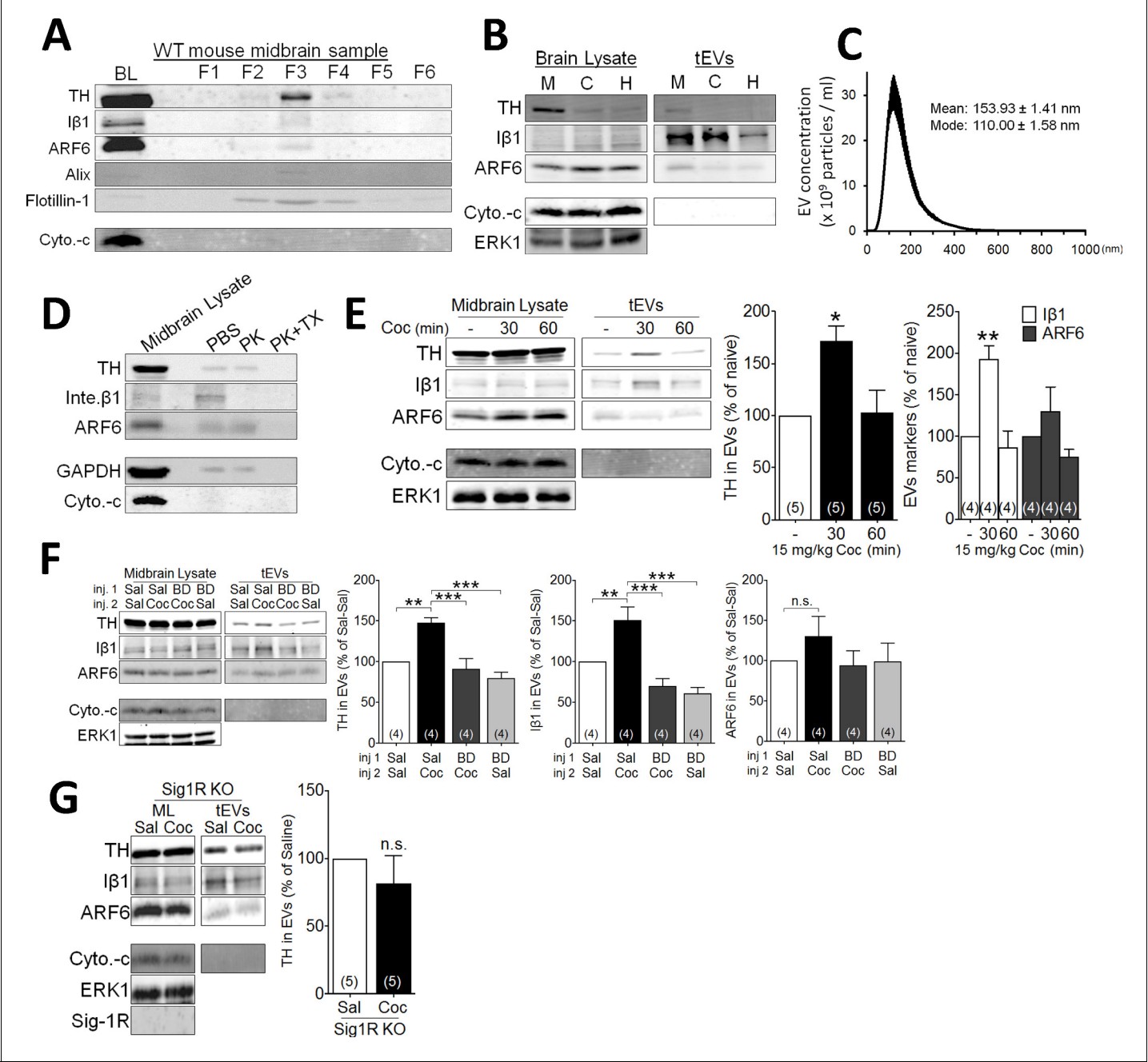

**Figure 2.** Effect of cocaine on EV secretion in mouse midbrain. (**A**) Representative western blots of different sucrose fractions (F1–F6) of EVs isolated from mouse midbrain, showing the markers tyrosine hydroxylase (TH), Iβ1 (Inte. β1), ARF6, Alix, and Flotillin-1. The mitochondrial marker (cytochrome-c: Cyto.-c) was also used as a control, and is western blots from total brain lysates (BL) are also shown (2–3 replicates). (**B**) Representative western blots from tEVs obtained from midbrain (**M**), cortex (**C**), and hippocampus (**H**) (two replicates). (**C**) The size distribution of tEVs in mouse midbrain, as measured by NanoSight particle tracking (n = 3 replicates). (**D**) Proteinase K (PK) treatment of EV preparations from mouse midbrain, with, and without Triton-X (TX) included (two replicates). (**E**) Effect of cocaine (15 mg/kg, i.p.) on EV markers in preparations from WT mouse midbrain at several 30- and 60 min time points. Bar graphs of the experiments described in E (mean ± S.E.M; TH: $F_{2,12}$ = 7.3, p=0.0084, one-way ANOVA, *=p < 0.05 compared with naive, Dunnett's multiple comparison test; Iβ1: $F_{2,9}$ = 15.2, p=0.001, one-way ANOVA, **=p < 0.01 compared with naive, Dunnett's multiple comparison test; ARF6: $F_{2,9}$ = 2.5, p=0.14, one-way ANOVA). (**F**) Effect of the Sig-1R antagonist (BD1063: BD, 10 mg/kg, s.c.) on cocaine-evoked tEV release in WT mouse midbrain, 30 min after the in vivo cocaine injection. The Bar graphs shows mean effects from these experiments (mean ± S.E.M, TH: $F_{3,12}$ = 14.2, p=0.0003, one-way ANOVA, **=p < 0.01, ***=p < 0.001, Dunnett's multiple comparison test; Iβ1: $F_{3,12}$ = 16.3, p=0.0002, one-way ANOVA, **=p < 0.01, ***=p < 0.001, Dunnett's multiple comparison test; ARF6: $F_{3,12}$ = 1.5, p=0.26, n.s., not significant one-way ANOVA). (**G**) The effect of cocaine on tEV release is absent in Sig-1R knock out mouse midbrain, 30 min after in vivo cocaine injection. The Bar graph shows the means from

*Figure 2 continued on next page*

*Figure 2 continued*

this experiment (n.s., not significant, unpaired t-test). The number of replications of each experiment is shown in parentheses for each group in the bar graphs. Details All statistical comparisons. See *Source data 1* for values used in statistical analyses. *Figure 3—figure supplement 1* shows specificity of the Sig-1R antibody. *Figure 2—figure supplement 1* shows the protocol for isolation of EVs from mouse midbrain.

DOI: https://doi.org/10.7554/eLife.47209.005

The following figure supplement is available for figure 2:

**Figure supplement 1.** Brain EV Isolation Experimental Flow Chart.

DOI: https://doi.org/10.7554/eLife.47209.006

with its location on the plasma membrane (*van Niel et al., 2018*; *EL Andaloussi et al., 2013*; *Muralidharan-Chari et al., 2009*; *Imjeti et al., 2017*). In contrast, all three proteins were degraded by PK in tEV preparations treated with Triton X (*Figure 2D*), suggesting that, unlike Iβ1, TH and ARF6 are located within EVs, rather than on their membranes.

Because they were found in EV-rich preparations of midbrain, TH, Iβ1, and ARF6 were used as markers to evaluate the effect of cocaine on tEVs. Like NG-108 cells, cocaine (15 mg/kg) increased Iβ1 (and TH) levels in midbrain tissue within 30 min of an intraperitoneal (i.p.) injection (*Figure 2E*), and this returned to control levels 60 min following cocaine treatment (*Figure 2E*). However, ARF6 levels were not significantly altered by cocaine (*Figure 2E*). As in NG-108 cells, the cocaine-stimulation of tEV markers in midbrain was also prevented by the Sig-1R antagonist, BD1063 (*Figure 2F*). Moreover, cocaine failed to increase any of the tEV markers (*Figure 2G*) in midbrain preparations from mice lacking the Sig-1R gene (*Sigmar1*), suggesting that Sig-1Rs are essential for cocaine-induced tEV release in mouse midbrain.

## The Sig-1R associates with the inactive form of ARF6 in mouse midbrain

To determine cellular locations of the Sig-1R we used immunofluorescence confocal microscopy in the mouse ventral midbrain. We found that Sig-1R (*Mavlyutov et al., 2016*) and TH fluorescence signals were colocalized (*Figure 3A*), and as TH is a marker for DA neurons in the ventral midbrain, the data suggest that Sig-1Rs are found in DA neurons. However, the Sig-1R signal was also found associated with the vesicular GABA transporter (vGAT), a marker of GABA neurons in the mouse ventral midbrain (*Figure 3A*). Therefore, the Sig-1R is likely expressed in both DA and GABA neurons in the midbrain. Immunofluorescence confocal microscopy also revealed co-localization of Sig-1R and ARF6 in TH-positive neurons in the mouse ventral midbrain (*Figure 3C*), and these proteins co-immunoprecipitated in midbrain samples from wild-type, but not Sig-1R knockout mice (*Figure 3D*). Also, the Sig-1R immunohistochemical signal was absent in Sig-1R knockout mice (*Figure 3—figure supplement 1*).

The subcellular distribution of ARF6 in the mouse midbrain was next compared with Sig-1Rs in a fractionation assay allowing detection of the MAM (*Figure 3B*), where Sig-1Rs are abundant (*Hayashi and Su, 2007*; *Lewis et al., 2016*). Both the Sig-1R and ARF6 were found in this MAM fraction (*Figure 3B*), but another ARF GTPase, ARF-1, was not detected (*Figure 3B*). Together, our results indicate that Sig-1Rs and ARF6 colocalize with GABA and DA neuron markers and are associated with the MAM in the mouse midbrain.

## Involvement of Sig-1Rs, ARF6, and myosin light chain kinase in cocaine-induced EV release

To determine whether, like in NG-108 cells, cocaine-stimulation of EV secretion occurred through Sig-1R- and ARF6-dependent mechanisms, we manipulated signaling by these proteins, followed by preparation of midbrain tEV fractions. We found that an injection of cocaine (15 mg/kg, i.p.) significantly attenuated the co-immunoprecipitation of ARF6 and Sig1R in the mouse midbrain (*Figure 4A*), and this was prevented by a preceding subcutaneous (s.c.) injection of the Sig-1R antagonist, BD1063 (10 mg/kg) (*Figure 4B*). This suggests that the cocaine facilitates activation of the Sig-1R, and this triggers Sig-1R dissociation from ARF6. Next, we determined whether in vivo cocaine treatment altered the intracellular localization of ARF6, using the MAM fractionation assay. We found that, unlike that observed in the P3 fraction where ARF6 levels remained unchanged, 10 min after cocaine injection the level of MAM-associated ARF6 was decreased (*Figure 4C*). Moreover,

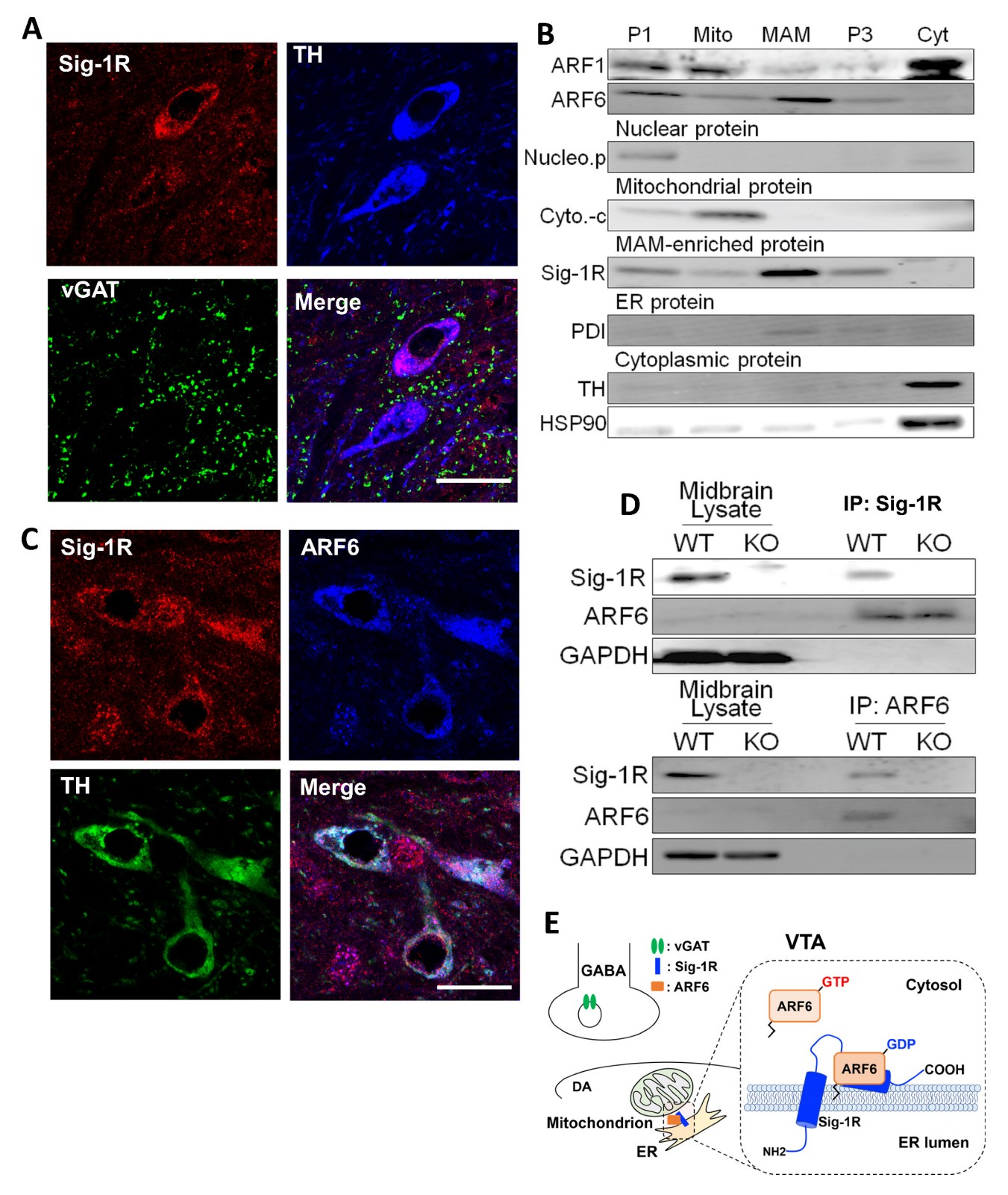

**Figure 3.** The Sig-1R interacts with ARF6 at the MAM in mouse midbrain. (**A**) Confocal microscopy shows Sig-1R fluorescence immunostaining (Red) in association with either TH (Blue)-, or vGAT (Green)-positive neurons in the wildtype mouse VTA. Scale bar = 50 μm. (**B**) The subcellular distribution of proteins in wildtype mouse midbrain (P1: nuclear fraction; Mito: mitochondrial fraction; P3: microsomal fraction, containing plasma membrane and ER; Cyt: cytosolic fraction; NucleoP: nucleoporin p62; Cyto-c: cytochrome-c; TH: tyrosine hydroxylase; HSP90: heat-shock protein 90). (**C**) Confocal

*Figure 3 continued on next page*

*Figure 3 continued*

microscopic images showing co-localization of fluorescence immunostaining of the Sig-1R (Red) and ARF6 (Blue) in TH (Green)-positive neurons in the wildtype mouse VTA (scale bar = 20 μm). (D) Immunoprecipitation (IP) of the Sig-1R/ARF6 complex. Brain lysates were prepared from wildtype or Sig-1R KO mouse midbrain, immunoprecipitated with anti-ARF6 antibody, and then probed with anti-Sig1R, ARF6, and GAPDH antibody. (E) Schematic drawing of the interaction between Sig1R and ARF6 in mouse midbrain. Each experiment was replicated twice. *Figure 3—figure supplement 1* shows the absence of Sig-1R immunofluorescence in the Sig-1R knockout mouse brain. Also see *Figure 8—figure supplement 1* for proposed interaction between the Sig-1R and ARF6.

DOI: https://doi.org/10.7554/eLife.47209.007

The following figure supplement is available for figure 3:

**Figure supplement 1.** Absence of Sig-1R immunofluorescence in Sig-1R knockout mouse brain.

DOI: https://doi.org/10.7554/eLife.47209.008

Sig-1R levels were not significantly altered in either the P3 or the MAM fractions (*Figure 4C*). These results suggest that the Sig-1R is activated by cocaine while associated with the MAM and this facilitates dissociation of the Sig-1R from ARF6. As ARF6-GTP modulation by the GEF inhibitor SH3 altered EV secretion in NG-108 cells (*Figure 1E*), we measured its effect (s.c., 10 mg/kg) on cocaine-stimulated tEV secretion in mouse midbrain. Consistent with NG-108 cell data, SH3 significantly inhibited the cocaine-induced increase of TH and Iβ1 in mouse midbrain (*Figure 4D*). Existing data also support the involvement of cytoskeletal myosin and actin in EV release and show that ARF6 exerts its effects on EV release through phosphorylation of myosin light-chain kinase (MLCK) (*van Niel et al., 2018*; *Muralidharan-Chari et al., 2009*). Therefore, we examined MLCK involvement in the cocaine-simulated EV release in midbrain tissue and found that the MLCK inhibitor ML7 (2 μM) prevented the increase in EV release, as measured by Iβ1, or TH in EV-rich fractions (*Figure 4D*).

In consideration of these data, we propose the following model; 1) the Sig-1R forms a stable complex with the inactive ARF6-GDP at the MAM, 2) cocaine, through interaction with the Sig-1R, causes dissociation of the ARF6-GDP/Sig-1R complex, 3) free ARF6-GDP is then converted to the active ARF6-GTP by GEFs, and 4) ARF6-GTP translocates to the plasma membrane where it stimulates EV release into the extracellular space (*Figure 4E*) by activating MLCK, and permitting EV mobility. Using this model of EV secretion, we next sought to determine its functional relevance to synaptic modulation by eCBs in the mouse midbrain.

## 2-AG is found in EV-enriched midbrain fractions

A recent study found that microvesicle-enriched fractions from primary microglia cultures contained the eCB anandamide (*Gabrielli et al., 2015a*), and work from our laboratory showed that cocaine promotes the release of eCB 2-AG in the midbrain (*Wang et al., 2015*). However, the potential involvement of EVs in 2-AG function has not been assessed. To determine whether 2-AG is found in EV fractions from mouse midbrain, we used Fourier transform mass spectrometry (FTMS). We found that the levels of 2-AG were higher in midbrain homogenates than in cerebral cortex, and were approximately fivefold larger than those observed in tEV fractions from these brain regions (*Figure 5A*). The concentration of 2-AG in midbrain tEV fractions (206.9 ± 70.2 pmol/mg, *Figure 5A*) was also higher than that measured in the cerebral cortex (121.4 ± 16.1 pmol/mg, *Figure 5A*), suggesting regional differences in concentrations of 2-AG. We also found that cocaine significantly increased 2-AG levels in midbrain tissue (*Figure 5B*). However, when cocaine-stimulation of 2-AG levels in tEV fraction were measured using FTMS in pooled samples of mouse midbrain, we observed considerable variability in baseline saline-injected controls (n = 15 mice in three experiments; *Figure 5C*), and in cocaine-stimulated levels of the eCB (n = 15 mice in three experiments). Thus, although a clear trend toward increased 2-AG in these tEV fractions was observed, and cocaine significantly increase midbrain tissue levels of 2-AG (*Figure 5B*), the effect of cocaine on 2-AG content in the tEV fractions was not significant ($t_8 = 1.61$, p=0.147, unpaired Student's t-test; *Figure 5C*).

Recent studies show that fatty acid binding proteins can act as intracellular carriers for 2-AG (*Kaczocha et al., 2009*), and one of these, fatty acid binding protein 5 (FABP5), was involved in mediating extracellular 2-AG release in the mouse brain (*Haj-Dahmane et al., 2018*). To determine whether this carrier of 2-AG could also be localized to midbrain EVs, we isolated EV fractions from mouse midbrain and used western blots to measure FABP5 and other EV markers. These EV

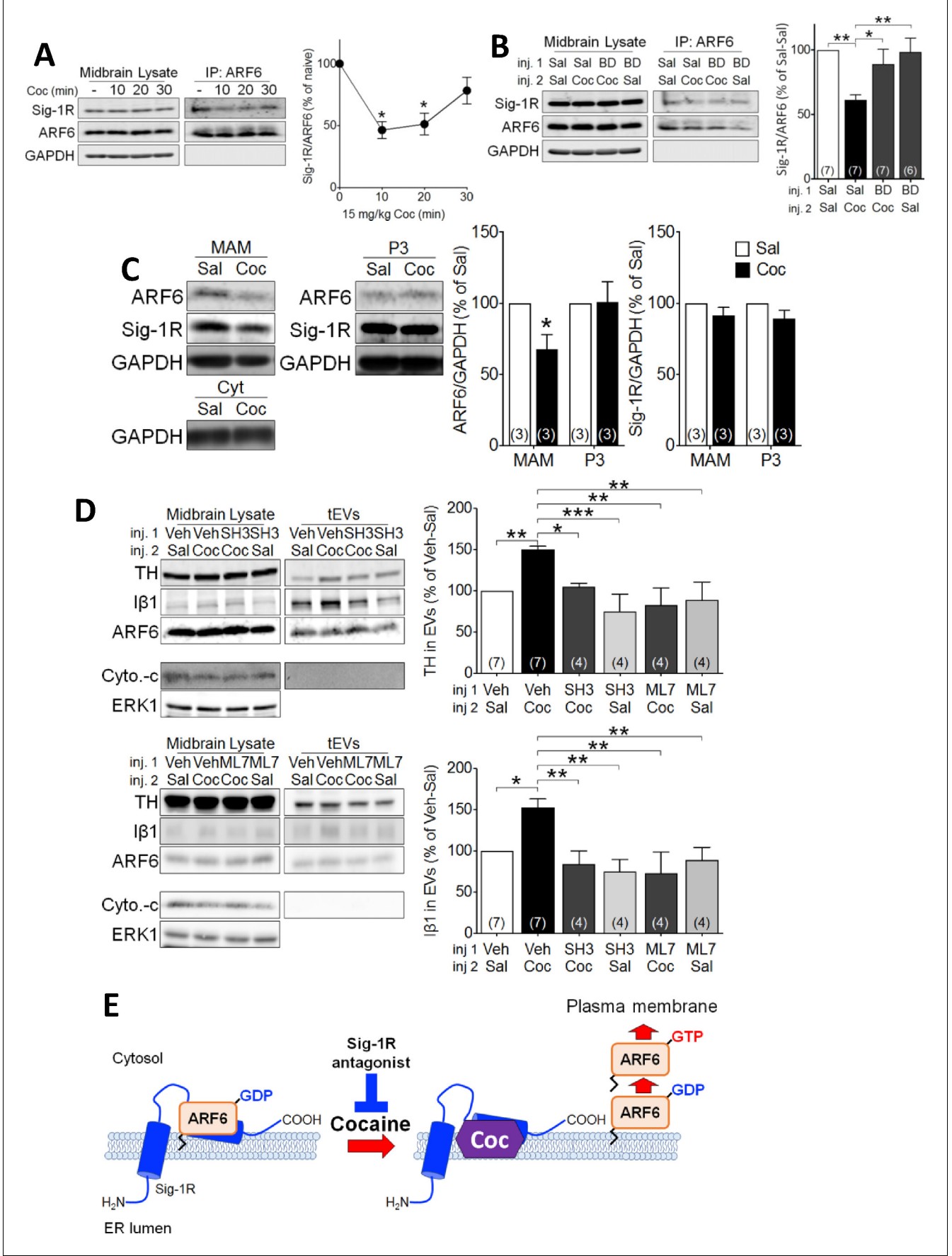

**Figure 4.** Cocaine causes translocation of ARF6 via its dissociation from the Sig-1R in mouse midbrain. (A) Western blots showing that cocaine reduces the interaction between ARF6 and the Sig-1R in a time-dependent manner in mouse midbrain. The graph shows mean (± s.e.m.) of co-IP of ARF6 and Sig-1R, before, and 10, 20 and 30 min after in vivo cocaine injection (n = 4; ($F_{3,12}$ = 4.3, p=0.028, one-way ANOVA, *=p < 0.05 compared with naive, Dunnett's multiple comparison test). *p<0.05, **p<0.01; one-way ANOVA followed by Dunnett post-hoc test). (B) Effect of the Sig-1R antagonist BD1063 (BD, 10 mg/kg, s.c.) on the dissociation of the ARF6-Sig-1R complex in mouse midbrain, 10 min after i.p. cocaine injection. BD1063 was injected 20 min before cocaine. The bar graph represent mean ± s.e.m. (n = 7; $F_{3,23}$ = 5.3, p=0.006, one-way ANOVA, *=p < 0.05, **=p < 0.01, Dunnett's multiple comparison test (C) Western blots showing the effect of cocaine versus saline injection on ARF6 concentration associated with the MAM, or P3 in mouse midbrain at 10 min post-i.p. injection. Bar graphs show mean (± S.E.M., n = 3) expression of ARF6 or Sig-1R as a proportion of GAPDH protein in MAM or P3 preparations, for all conditions, expressed as the percent response observed following saline injection (*=p < 0.001, unpaired t-test). (D) Western blots showing the effect of the ARF6 GEF inhibitor (SecinH3: SH3, 10 μmol/kg, s.c.) or the MLCK inhibitor, ML7 (2 μM, s.c.) on cocaine-evoked EV marker release in mouse midbrain, 30 min after i.p. cocaine or saline injection. SH3, ML7, or vehicle was injected 20 min prior to cocaine or saline injection. ERK1 is used as a control protein. The bar graphs represent the mean (± S.E.M) concentration of TH or Iβ1 expressed as a percentage of the level seen following vehicle-saline control injections(n = 4–7, TH: $F_{3,12}$ = 7.9, p=0.004, one-way ANOVA, *=p < 0.05, **=p < 0.01, Dunnett's multiple comparison test; Iβ1: $F_{3,12}$ = 7.0, p=0.006, one-way ANOVA, *=p < 0.05, **=p < 0.01, Dunnett's multiple comparison test). (E) Schematic illustrating of the effect of cocaine on the Sig1R-ARF6 interaction in mouse midbrain. See *Source data 1* for values used in statistical analyses. Also see *Figure 8—figure supplement 1* for proposed interaction between the Sig-1R and ARF6 and cocaine.

DOI: https://doi.org/10.7554/eLife.47209.009

fractions contained FABP5 as well as the EV markers TH, Iβ1, and flotillin-1 (*Figure 5D*). This suggests that the FABP5 protein is associated with EVs to perhaps mediate 2-AG signaling in the CNS.

## Sig-1R antagonism prevents cocaine-stimulated synaptic 2-AG function in VTA DA neurons

There is strong evidence that 2-AG is synthesized in rodent midbrain VTA neurons, where it can modulate synaptic neurotransmitter release (*Riegel and Lupica, 2004*; *Melis et al., 2004*; *Parsons and Hurd, 2015*; *Labouèbe et al., 2013*). Moreover, 2-AG function is increased during heightened DA neuron activity (*Riegel and Lupica, 2004*; *Melis et al., 2004*), or when phospholipases are activated by certain $G\alpha_{q11}$-containing GPCRs, such as the $\alpha_1$-noradrenergic ($\alpha_1$R), or type-I metabotropic glutamate receptors (mGluRIs) (*Wang et al., 2015*; *Haj-Dahmane and Shen, 2014*). These data also show that cocaine's ability to increase VTA 2-AG function occurs via its inhibition of the norepinephrine transporter (NET), causing activation of $\alpha_1$Rs on VTA DA neurons and 2-AG synthesis from membrane phospholipids (*Wang et al., 2015*). Based on this previous work, and our data showing cocaine interactions with midbrain Sig-1Rs, ARF6 and EV release, we evaluated the possibility that 2-AG function in the VTA occurs via EV- and Sig-1R-dependent mechanisms in mouse midbrain DA neurons.

Local 2-AG function can be measured with high temporal fidelity through its activation of CB1Rs leading to local inhibition of synaptic transmission (*Alger, 2002*). This functionally relevant endogenous 2-AG reduces inhibitory postsynaptic currents (IPSCs) mediated by synaptic GABA release onto $GABA_B$ receptors ($GABA_B$Rs) located on DA neuron dendrites (*Wang et al., 2015*; *Riegel and Lupica, 2004*). Similar to previous data from rat VTA DA neurons (*Wang et al., 2015*), we found that cocaine (10 μM) inhibited IPSCs recorded in mouse DA neurons (*Figure 5E and F*). The IPSC inhibition by cocaine was prevented by the CB1R antagonist, AM251 (1 μM; *Figure 5H-Figure 5—figure supplement 1*) and was absent in mice lacking the CB1R (*Zimmer et al., 1999*) (*Figure 5E and F*). The inhibition of IPSCs by cocaine was also reduced by tetrahydrolipostatin (THL, 2 μM), an inhibitor of the enzyme diacylglycerol lipase-α (DGLα), preventing conversion of diacylglycerol (DAG) to 2-AG (*Figure 5—figure supplement 1A1*, *Figure 5—figure supplement 1B*). Cocaine-mediated 2-AG release was also absent in mutant mice lacking expression of DGLα in DA neurons (*Shonesy et al., 2014*) (*Dagla^{flox/flox}* x *DAT^{Cre}* mice; *Figure 5G and H*). These experiments confirm that inhibition of GABA release onto DA neurons by cocaine occurs via stimulation of 2-AG function in the mouse VTA.

We next examined Sig1-R involvement in cocaine-dependent 2-AG release in mouse VTA DA neurons. Each of two Sig-1R antagonists (BD1063 or NE100; 2 μM) significantly reduced the cocaine (10 μM) simulation of 2-AG release in VTA DA neurons (*Figure 6A–C and E*). This effect of cocaine was also significantly reduced in DA neurons from Sig-1R knockout mice, particularly 5–10 min after beginning cocaine application (*Figure 6D and E*). Importantly, the inhibition of IPSCs by the synthetic CB1R agonist, WIN55,212–2 (1 μM), was not reduced by Sig-1R antagonism, or by genetic

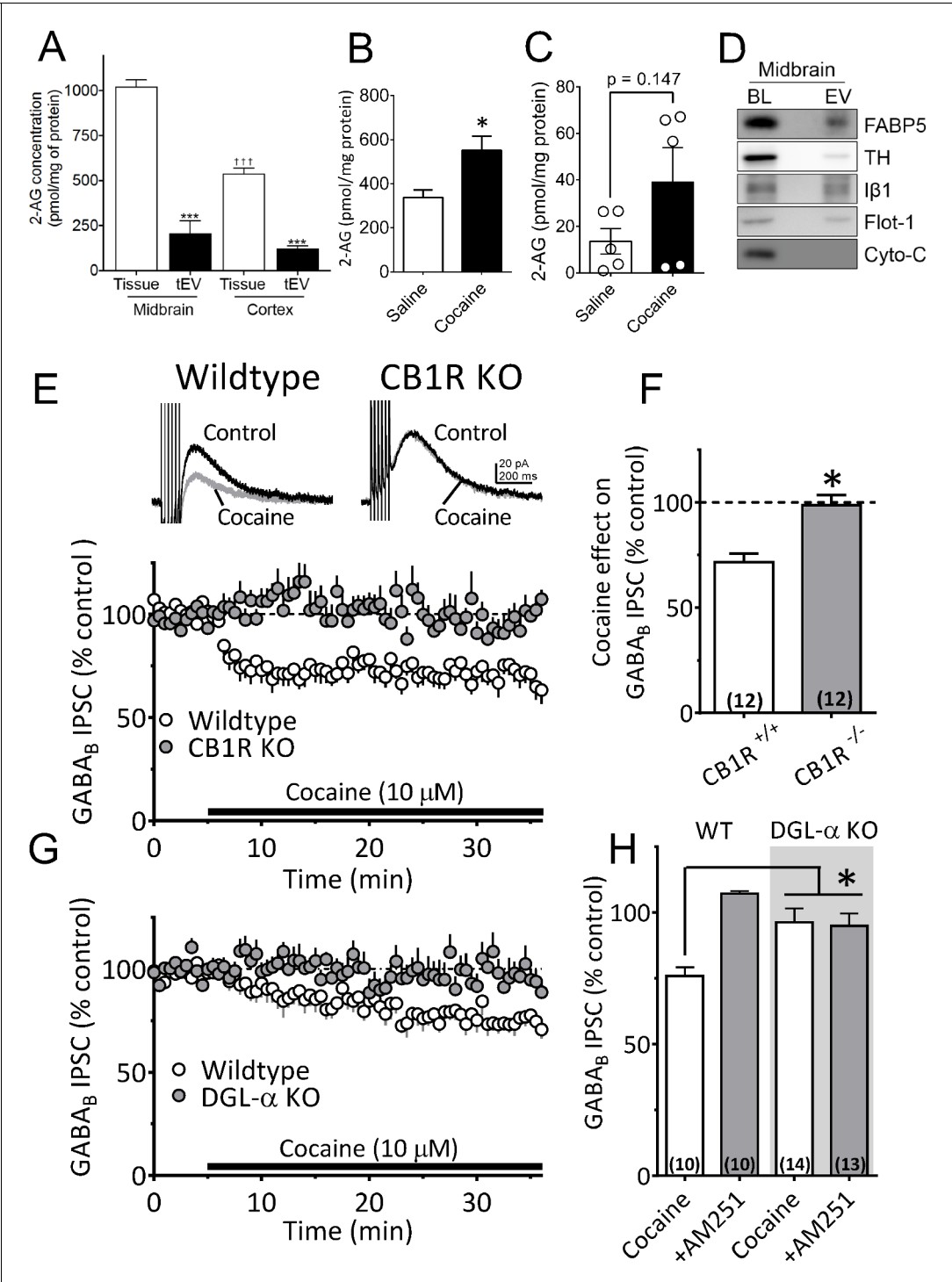

**Figure 5.** Cocaine-stimulation of 2-AG accumulation in midbrain tEVs and brain slices. (**A**) Levels of 2-AG measured in midbrain and cortex tissue homogenates and in tEVs from these same brain regions using Fourier transform mass spectrometry (FTMS; $F_{3,8}$ = 86.92, p<0.0001; Tukey's posthoc test, ***=p < 0.05, †††=p < 0.05, tissue midbrain vs. cortex, n = 3). (**B**) Comparison of the concentration of 2-AG in midbrain homogenates from mice injected with saline or cocaine 15 min prior to dissection (mean ± S.E.M.; *=p < 0.05, unpaired Student's t-test, n = 3). (**C**) Levels of 2-AG measured using FTMS in fr3 containing tEVs isolated from mouse midbrain 15 min after in vivo injection with saline or 10 mg/kg cocaine. Each point represents data pooled from three mice ($t_8$ = 1.61, p=0.147, unpaired Student's t-test; n = 15 mice per group). (**D**) Western blots detecting fatty acid binding protein-5 (FABP5), TH, Iβ1, Flot-1 and Cyto-C in either whole brain lysate (BL) or in the EV fraction (fr3) obtained via sequential centrifugation and sucrose-gradient separation. Note that all EV marker proteins are detected in the BL preparation and that FABP5 is also found in this EV fraction. (**E**) Cocaine stimulates 2-AG inhibition of GABA release onto VTA DA neurons in vitro. Cocaine application inhibits GABA<sub>B</sub>-receptor-mediated synaptic

*Figure 5 continued on next page*

*Figure 5 continued*

IPSCs in DA neurons from wildtype mice, but not in CB1R knockout (KO) mice. (**F**) Mean inhibition by cocaine of IPSCs in wildtype and CB1R-KO mice (p=0.0004, unpaired t-test). (**G**) The inhibition of IPSCs by cocaine is absent in mice lacking the gene (*Dagla*) encoding the 2-AG synthetic enzyme, DGL-α, in DA neurons. (**H**) Mean effects of cocaine on IPSCs in the presence and absence of the CB1R antagonist/inverse agonist (AM251, 4 μM) in wildtype and DGL-α-KO mice. Note the reversal of the cocaine inhibition by AM251 in wildtype DA neurons, the absence of inhibition of IPSCs by cocaine, and lack of effect of AM251 in the neurons from DGL-α-KO mice ($F_{3, 40}$ = 8.3, p=0.0002, one-way ANOVA, p=0.009, Tukey's multiple comparison post-hoc test). *Figure 5—figure supplement 1* shows that blockade of CB1Rs or 2-AG synthesis also prevents inhibition of IPSCs by cocaine. See *Source data 1* for values used in statistical analyses.

DOI: https://doi.org/10.7554/eLife.47209.010

The following figure supplement is available for figure 5:

**Figure supplement 1.** Blockade of CB1Rs or 2-AG synthesis prevents inhibition of IPSCs by cocaine.

DOI: https://doi.org/10.7554/eLife.47209.011

deletion of this receptor (*Figure 6—figure supplement 1*). This indicates that Sig-1Rs are linked to cocaine-stimulated 2-AG function in the CNS, and that CB1R signaling is not diminished by altered Sig-1R function or expression.

To examine whether Sig-1Rs are involved in facilitating 2-AG release derived from direct GPCR activation, we determined whether $α_1$R and mGluRI co-activation could stimulate 2-AG function in mouse VTA, and whether this is altered in Sig-1R knockout mice. Consistent with our previous report (*Wang et al., 2015*), co-application of the $α_1$R agonist phenylephrine (PE, 100 μM) and the mGluRI agonist, DHPG (1 μM) inhibited GABA_B IPSCs in wildtype mouse VTA DA neurons, and this was blocked by AM251 (*Figure 7B and C*). However, it is also important to note that the properties of the IPSC inhibition produced by DHPG+PE differed from that seen with cocaine. Thus, the response to DHPG+PE was much slower to reach maximum and lacked the early fast component observed with cocaine (*Figure 7—figure supplement 1*) in wildtype mice. Therefore, in comparison, the effect of DHPG+PE primarily consisted of the delayed slow component (*Figure 7—figure supplement 1C*). Also, in DA neurons from Sig-1R knockout mice, the slow response to DHPG+PE was significantly smaller (*Figure 7A–7C*, *Figure 7—figure supplement 1A*), which contrasts with that seen with cocaine where the early fast inhibition was absent, but the later inhibition was less affected in Sig-1R knockout mice (*Figure 6D*, *Figure 7—figure supplement 1B*). These differences could indicate reliance upon distinct signaling pathways that convergence upon Sig-1Rs to permit 2-AG release via EVs.

To determine whether the effects of 2-AG derived from a non-GPCR source are also altered in the Sig-1R knockout mouse, we measured tonic 2-AG release that is observed without GPCR activation (either indirectly by cocaine or directly by DHPG+PE) (*Wang et al., 2015*). The tonic inhibition of GABA_B IPSCs mediated by this basal level of endogenous 2-AG is revealed when CB1Rs are blocked by antagonists, resulting in an increase in these synaptic currents (*Wang et al., 2015*; *Riegel and Lupica, 2004*). We found that DA neurons from both wildtype and Sig-1R knockout mice exhibited similar significant IPSC increases when the CB1R antagonist AM251 was applied (*Figure 7A and B*). Therefore, the data suggest that only 2-AG derived from GPCR stimulation is dependent upon intact Sig-1R function, and additionally that 2-AG synthesis itself is not disrupted in Sig-1R knockout mice.

Our NG-108 experiments indicated that Sig-1Rs stabilize the inactive GDP-bound form of ARF6, and that cocaine activates GTP-bound ARF6 through an interaction with Sig-1Rs, thereby permitting EV release. Moreover, our FTMS experiments identified 2-AG in midbrain tEV fractions (*Figure 5A*, *Figure 5C*). Therefore, involvement of ARF6 in the 2-AG-dependent inhibition of GABA release by cocaine was tested in wild-type mouse VTA DA neurons. Manipulation of ARF6 activation with the GEF inhibitor, SH3 (*Figure 8A and E*), or, direct inhibition of ARF6 with NAV2729 (both at 10 μM) (*Yoo et al., 2016*), significantly inhibited cocaine-induced 2-AG function in midbrain DA neurons (*Figure 8B and E*). Also, like that observed with Sig-1R antagonists or knockouts (*Figure 6*), the reduction in the cocaine inhibition of IPSCs by both SH3 and NAV2729 was more prominent within the first 10 min of cocaine application (*Figure 8A*, *Figure 8B*). As inhibition of MLCK significantly reduced EV release in midbrain tissue experiments, we examined its involvement in the synaptic effects of cocaine-simulated 2-AG function in DA neurons. We found that the MLCK inhibition by

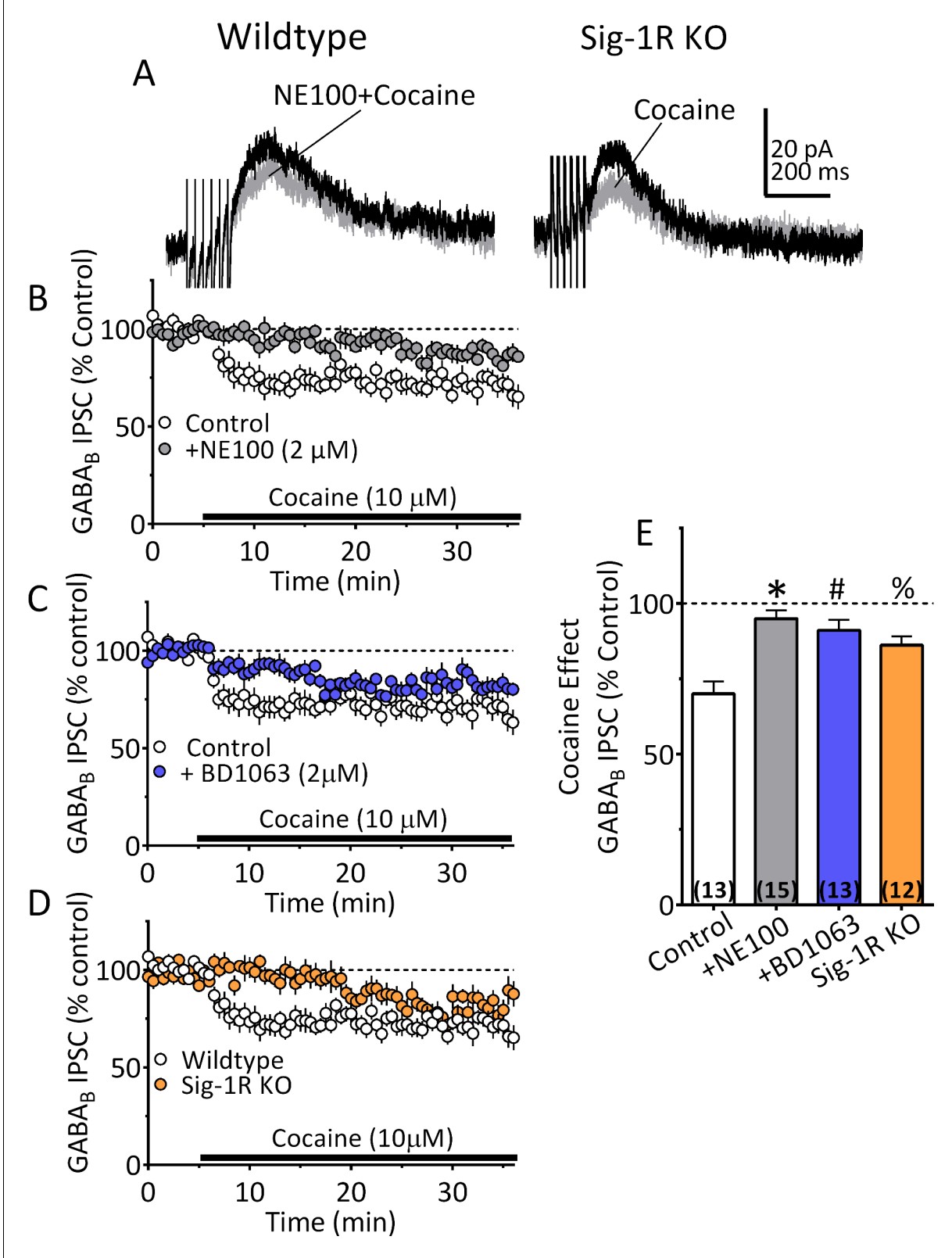

**Figure 6.** Inhibition of IPSCs by cocaine in VTA DA neurons depends upon Sig-1Rs. (A) Mean waveforms showing the effect of cocaine (10 μM) on GABA$_B$ IPSCs in a DA neuron from a wildtype mouse during application of the Sig-1R antagonist NE100 (2 μM, left), or in a cell from a Sig-1R KO mouse (right). (B) Mean time-course showing effect of cocaine on IPSCs in absence (Control) and presence of NE100 in wildtype mice. (C) Mean time-course showing effect of cocaine on IPSCs in absence (Control) and presence of BD1063 (2 μM) in wildtype mice. (D) Time-course of cocaine effects in

*Figure 6 continued on next page*

*Figure 6 continued*

wildtype and Sig-1R KO mice. (**E**) Summary of Data shown in A-D. The inhibition of IPSCs by cocaine was significantly reduced by NE100 or BD1063 in wildtype mice and was significantly smaller in Sig-1R KO mice ($F_{3,49}$ = 10.90, one-way ANOVA, p<0.0001; *=p < 0.0001, #=p = 0.0002, %=p = 0.005, Dunnett's multiple comparisons post-hoc test. *Figure 6—figure supplement 1* shows that antagonism or knockout of the Sig-1R does not change CB1R function in mouse VTA DA neurons. See *Source data 1* for values used in statistical analyses.

DOI: https://doi.org/10.7554/eLife.47209.012

The following figure supplement is available for figure 6:

**Figure supplement 1.** Antagonism or knockout of the Sig-1R does not reduce CB1R function in mouse VTA DA neurons.
DOI: https://doi.org/10.7554/eLife.47209.013

ML7 (2 µM) also significantly reduced the effect of cocaine on 2-AG release in this electrophysiological assay of eCB function (*Figure 8C and E*).

Together these data demonstrate that EV release is controlled by the Sig-1R, ARF6, and MLCK, and that cocaine's interaction with the Sig-1R can recruit this signaling cascade. The data further demonstrate that disruption of these signaling mechanisms leads to reduced synaptic 2-AG function in the midbrain, thereby implicating these proteins and EVs in the release of eCBs.

## Discussion

Previous studies show that a cocaine binds to Sig-1Rs (*Sharkey et al., 1988*; *Chen et al., 2007*; *Hiranita et al., 2011*), and that blockade of this interaction reduces effects of the psychostimulant (*Romieu et al., 2002*; *Hiranita et al., 2011*; *Lever et al., 2014*; *Fritz et al., 2011*). Additionally, cocaine's actions at Sig1-Rs alters its ability to influence voltage-gated potassium channel function, and this can reduce its behavioral effects (*Kourrich et al., 2013*; *Romieu et al., 2002*; *Lever et al., 2014*; *Fritz et al., 2011*). The present data demonstrate that the Sig-1R also regulates EV secretion in cultured cells and in the mouse midbrain, and that cocaine modulates this process through interaction with the Sig-1R. We also show that the interactions among Sig1-Rs, cocaine, and EVs can regulate synaptic transmission in the brain via the control of 2-AG release and its inhibition of GABAergic input to DA neurons in the mouse VTA. Therefore, our study identifies novel mechanisms for Sig-1R control of EV function and implicates EVs in eCB release in the CNS.

EVs are increasingly recognized as a highly regulated mechanism to permit exchange of signaling molecules, such as lipids, nucleic acids, organelles, and proteins, among cells (*van Niel et al., 2018*). As such, regulatory control points for EV formation, budding, translocation, and cargo release have been delineated in many cell types during normal cellular function, and in disease states (*van Niel et al., 2018*; *Huang-Doran et al., 2017*; *EL Andaloussi et al., 2013*; *Muralidharan-Chari et al., 2009*; *Wang et al., 2017*; *Yoo et al., 2016*). Here, we show that cocaine treatment of NG108 cells, or of mouse midbrain after in vivo injection, stimulates EV release, and that this is mimicked by agonists of Sig-1Rs, and prevented by antagonists or genetic elimination of these receptors. Moreover, using co-immunoprecipitation assays, we provide evidence for an association between ARF6, an established regulator of EV secretion (*D'Souza-Schorey and Chavrier, 2006*; *Yoo et al., 2016*), and the Sig-1R in TH-positive VTA neurons, and find that blockade of ARF6 activation prevents cocaine-induced EV release in both NG-108 cells and midbrain. We also report that in vivo cocaine causes the ARF6/Sig1R complex to dissociate, and this is prevented by Sig-1R antagonism. These data suggest that Sig-1Rs bind ARF6 proteins to hold them in an inactive GDP-bound form, and that cocaine facilitates the dissociation of these proteins to permit conversion of ARF-GDP to the active ARF6-GTP. Our data also suggest that this interaction between ARF6 and Sig-1Rs occurs at the MAM, and that cocaine enables translocation of ARF6-GTP to the plasma membrane. This mechanism is notable because ARF6 is implicated in EV secretion via regulation of cytoskeletal actin function in a wide range of mammalian tissues (*D'Souza-Schorey and Chavrier, 2006*; *Yoo et al., 2016*), and this is supported by our observation that inhibition of MLCK also prevents the cocaine-induced increase in EV levels in mouse midbrain.

Previous work shows that anandamide is found in EV-containing membrane fractions of rodent microglia cultures, and that these fractions exhibit cannabinoid agonist properties when applied to hippocampal brain slices (*Gabrielli et al., 2015a*). Here, we show using FTMS that 2-AG is found in acute mouse midbrain preparations that are enriched in tEVs, and that 2-AG levels are significantly

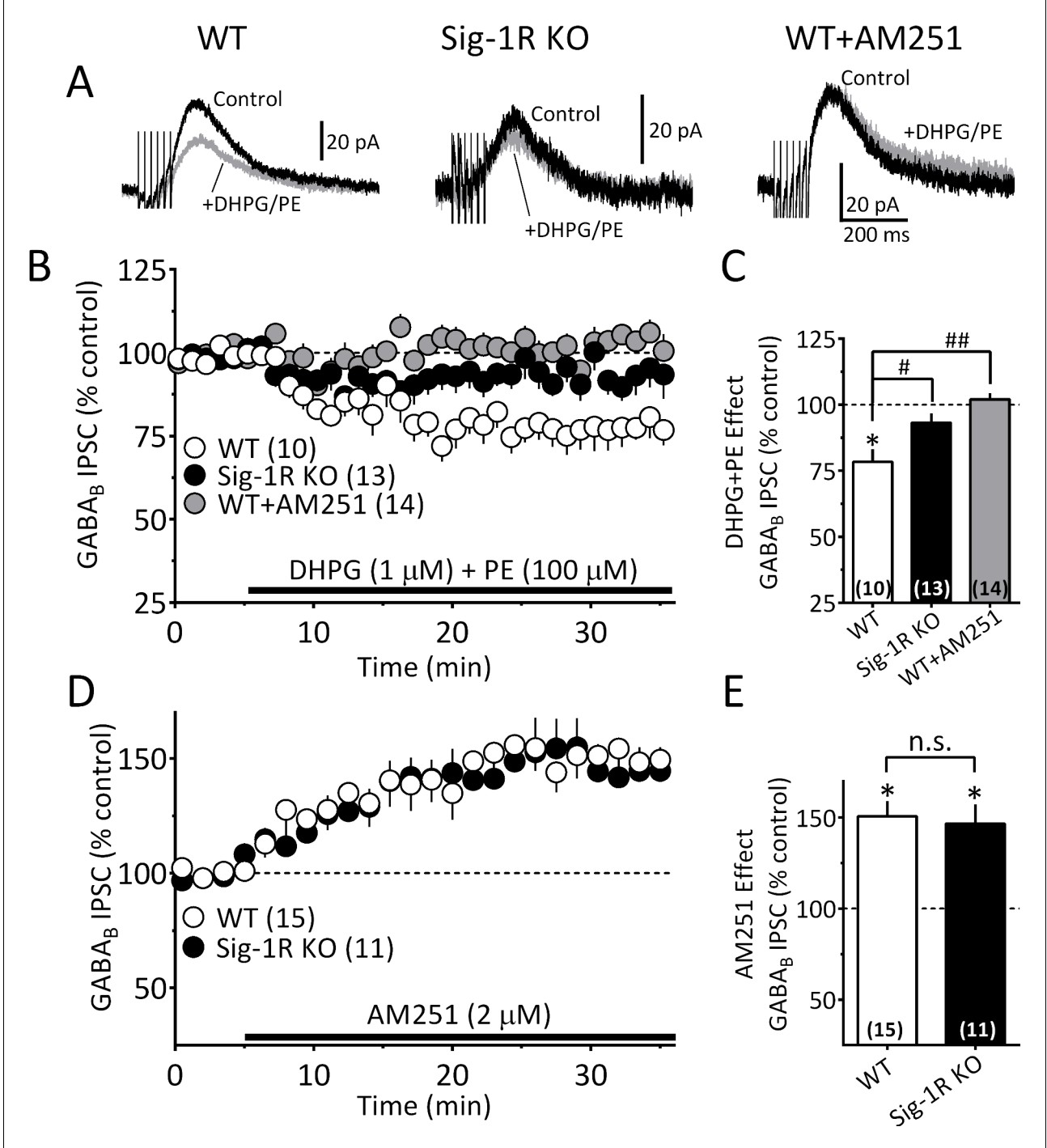

**Figure 7.** The Sig-1R is necessary for GPCR-induced but not tonic 2-AG release in the mouse VTA. (A) Mean GABA$_B$ IPSC waveforms collected during baseline (control, black line) and during co-application of DHPG and PE (gray line), in DA neurons from wildtype (WT, left), and Sig-1R knockout (KO, center) mice. Also shown is the effect of DHPG+PE in a representative neuron from a WT mouse following preincubation with AM251 (right). (B) Mean time courses of the effects of DHPG+PE in DA neurons from WT, sig-1R KO mice, and WT mice that had been pre-treated with AM251. The effect of DHPG+PE was significant (one-way repeated measures ANOVA, F$_{1.5, 110}$ = 133, p<0.0001), and this was significantly reduced in the Sig-1R KO, and by AM251 (Tukey's post hoc test p<0.0001). (C) Bar graph of data from the last 5 min of application of DGPG+PE as shown in B. The inhibition of IPSCs by DHPG+PE was significant (t$_9$ = 4.5, *=p = 0.0014, and the this was significantly reduced in the Sig-1R KO and AM251 groups (F$_{2,34}$ = 11.0, p=0.0002, one-way ANOVA; ##=p < 0.0001; #=p = 0.0013, Dunnett's posthoc test, the number of cells in each condition is indicated in parentheses). (D) Mean time course showing tonic inhibition of GABA$_B$ IPSCs by endogenous 2-AG, as revealed by antagonist of CB1Rs with AM251 in neurons from wildtype (WT) and Sig-1R KO mice (n = 15 and 11, respectively). (E) Bar graph of the change in IPSC amplitude during the last 5 min of AM251 application for

*Figure 7 continued on next page*

*Figure 7 continued*

data shown in D. AM251 caused a significant increase in mean IPSC amplitude in both groups (two-tailed unpaired t-test; \*\*=p < 0.0001, \*=p = 0.001), but there was no significant difference in this effect between groups (n.s. = not significant, two-tailed unpaired t-test, p=0.76). These data show that Sig-1Rs are necessary for the GPCR-induced 2-AG release caused by DHPG+PE (**A–C**), but not for tonic non-GPCR-dependent 2-AG release (**D–E**), and they suggest that DGLα function is not impaired in Sig-1R KO mice. *Figure 7—figure supplement 1* shows kinetic differences between 2-AG function elicited by DHPG+PE and cocaine in the mouse VTA. See *Source data 1* for values used in statistical analyses.

DOI: https://doi.org/10.7554/eLife.47209.014

The following figure supplement is available for figure 7:

**Figure supplement 1.** Differences between 2-AG function elicited by DHPG+PE and cocaine in the mouse VTA.

DOI: https://doi.org/10.7554/eLife.47209.015

increased in midbrain homogenates after in vivo exposure to cocaine. In contrast, although 2-AG could be measured in tEV fractions using FTMS in mouse midbrain, and tEV markers were significantly increased after in vivo cocaine treatment, the increase in 2-AG levels produced by cocaine in the tEV preparation did not reach statistical significance despite a clear trend. As these preparations are technically demanding and yield small amounts of material, it is possible that the between-groups ex vivo design and variability among samples in both saline control and cocaine injected mice contributed to this outcome. Alternatively, it is possible that cocaine causes an increase in 2-AG-containing EV release, but that the amount of 2-AG per vesicle does not change, and this increase in vesicle release could be sufficient to locally activate CB1Rs on GABAergic axon terminals.

The observation that cocaine increased midbrain levels of 2-AG provides biochemical support for our finding of cocaine-increased 2-AG function in mouse (this study) and rat VTA DA neurons in vitro (*Wang et al., 2015*). In this regard, we demonstrate that cocaine stimulates a 2-AG-dependent inhibition of GABA$_B$ receptor-mediated synaptic responses that is absent in mice lacking the CB1R, or the 2-AG biosynthetic enzyme, DGLα, in DA neurons. Based upon present data and our published work (*Wang et al., 2015*), we propose that 2-AG synthesis is stimulated when cocaine blocks norepinephrine uptake in the VTA, resulting in activation of G-protein-α$_q$-coupled α$_1$Rs, which, together with G$_q$-coupled mGluRIs stimulated by endogenous glutamate, activate phospholipases and liberate 2-AG from precursor membrane lipids (*Figure 8—figure supplement 1*) (*Kano et al., 2009*; *Maejima et al., 2005*; *Alger and Kim, 2011*; *Wang et al., 2015*; *Haj-Dahmane and Shen, 2014*; *Mátyás et al., 2008*). Although this model of 2-AG synthesis is supported by our studies, the mechanism of 2-AG is release is unknown. Here, using this 2-AG-sensitive synaptic response, we find that the same manipulations that blocked EV release in NG-108 cells and midbrain EV assays also reduced or eliminated cocaine-stimulated 2-AG effects on synaptic transmission in the mouse VTA. These manipulations include the disruption of Sig-1R signaling, the inhibition of ARF6 function, and the inhibition of MLCK. Moreover, we also found that the IPSC inhibition produced by a synthetic CB1R agonist was not altered by antagonism or genetic deletion of Sig-1Rs, suggesting that Sig-1Rs regulate 2-AG signaling but not CB1R function.

The involvement of Sig-1Rs in the GPCR-dependent 2-AG release was supported by experiments showing that co-activation of mGluRIs and α$_1$Rs by DHPG+PE could increase the release of this eCB, and that this was significantly reduced in Sig-1R KO mice. Moreover, another form of tonic 2-AG release that occurs under basal conditions in the absence of GPCR stimulation was unaltered in Sig-1R KO mice. Therefore, the data suggest that Sig-1Rs and EVs mediate only GPCR-dependent 2-AG release, and not that generated by other cellular pathways.

Based on our biochemical and electrophysiological data, we propose a model (*Figure 8—figure supplement 1*) in which cocaine initiates 2-AG synthesis via inhibition of the NET, leading to activation of α$_1$Rs coupled to G$_q$ proteins controlling phospholipases and the liberation of the 2-AG precursor DAG. DAG is then converted to 2-AG via DGLα and then packaged in EVs through an unknown process. 2-AG release from EVs is triggered when cocaine binds to Sig-1Rs to liberate ARF6-GDP and permit its conversion to the active ARF6-GTP, which can then act at MLCK to initiate EV fusion with the cellular membrane and release of 2-AG. Although these mechanisms are supported by the present data, our finding that the inhibition of IPSCs by 2-AG release by DHPG+PE is absent cells from Sig-1R KO mice suggests that cocaine binding to the Sig-1R is not necessary to initiate EV release. However, fundamental differences in the characteristics of the inhibition produced by these methods were noted. Thus, the kinetics of the 2-AG-mediated inhibition of GABA release

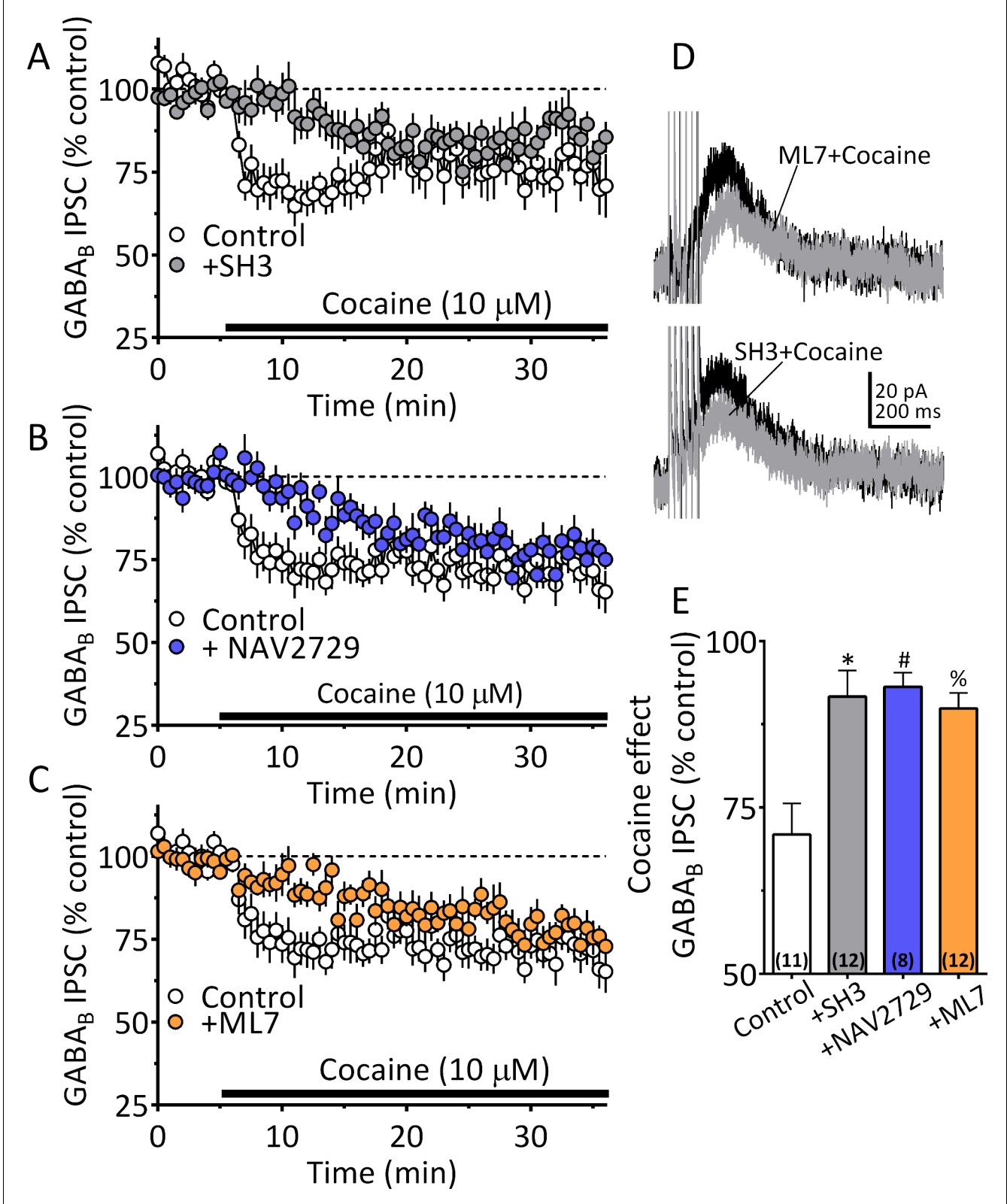

**Figure 8.** Cocaine stimulated 2-AG inhibition of GABA release is blocked by ARF6 inhibitors or myosin-light chain kinase (MLCK) inhibition. (**A**) Mean time-course of the effect of cocaine on GABA$_B$ IPSCs under control conditions, and during incubation with the ARF6 GEF inhibitor SH3 (10 µM). (**B**) Mean time-course of the effect of cocaine on the GABA$_B$ IPSCs under control conditions and during incubation with direct ARF6 inhibitor NAV2729 (10 µM). (**C**) Mean time-course of the effect of cocaine on the GABA$_B$ IPSCs under control conditions and during incubation with the MLCK inhibitor ML7 (2

*Figure 8 continued on next page*

Figure 8 continued

µM). (D) Mean waveforms of GABA$_B$ receptor-mediated IPSCs after addition of cocaine in cells preincubated with ML7 or SH3. (E) Summary of data with ML7, SH3, and NAV2729, shown in A-C. The effect of cocaine is significantly reduced by SH3, NAV2729, and ML7 (F$_{3,39}$ = 8.7, p=0.0002, one-way ANOVA, **=p < 0.001, Dunnett's multiple comparison test, *=p = 0.0003, #=p = 0.0005, %=p = 0.001; n for each condition shown in parentheses).
*Figure 8—figure supplement 1* shows our model of the proposed mechanisms underlying the cocaine-regulated synthesis and release of 2-AG in VTA DA neurons and the involvement of EVs and Sig1R-ARF6 signaling pathway See *Source data 1* for values used in statistical analyses.
DOI: https://doi.org/10.7554/eLife.47209.016
The following figure supplement is available for figure 8:

**Figure supplement 1.** Proposed mechanisms underlying the cocaine-regulated synthesis and release of 2-AG in VTA.
DOI: https://doi.org/10.7554/eLife.47209.017

caused by cocaine differ from DHPG+PE in that the effect onset and the peak response to cocaine occurred more rapidly than that seen with DHPG+PE (*Figure 6—figure supplement 1*). Also, the cocaine effect reached a maximum within approximately the first 5 min after application, and this early phase was completely blocked when Sig-1R, ARF6 or MLCK function was disrupted (*Figure 8*), whereas the smaller late phase of inhibition was resistant to these manipulations (*Figure 8*, *Figure 7—figure supplement 1*). Despite this, data showing that both the early and late phases of cocaine inhibition are prevented by AM251 (*Figure 5—figure supplement 1*) and absent in mice lacking the CB1R or DGLα (*Figure 5E–H*), indicate that both inhibitory phases depend upon 2-AG and CB1Rs. In contrast to the effect of cocaine, DHPG+PE does not produce a robust early phase of IPSC inhibition (*Figure 6—figure supplement 1*) and the delayed inhibition produced by the agonists is smaller, but not absent in Sig-1R KO mice (*Figure 7—figure supplement 1A*). These differences suggest that although cocaine and DHPG+PE initiate 2-AG-dependent inhibition of synaptic GABA release, they may involve distinct upstream mechanisms that converge on Sig-1Rs and their control of EV release. Thus, the faster time-course of the cocaine effect may result from its direct binding to Sig-1Rs (*Sharkey et al., 1988*; *Chen et al., 2007*; *Hiranita et al., 2011*) to more rapidly stimulate EV release, resulting in their depletion during the late phase. In contrast, the slower and more sustained effect of DHPG+PE on 2-AG release may reflect coupling of EV release to a signaling pathway that relies upon intracellular release of an endogenous Sig-1R agonist. In support of this, several putative endogenous Sig-1R agonists have been identified (*Monnet and Maurice, 2006*; *Ramachandran et al., 2009*; *Fontanilla et al., 2009*), and a more recent study shows that agonists of G$_q$-coupled receptors that stimulate phospholipases can increase intracellular levels of choline, which then acts as an agonist at Sig-1Rs to enhance their calcium signaling properties (*Brailoiu et al., 2019*). Therefore, we speculate that the distinct phases of 2-AG-dependent inhibition are related the ability of the cocaine to act as a direct agonist at Sig-1Rs, compared to potential indirect effects of DHPG+PE that may be mediated by an intracellular signaling molecule having agonist properties at sig-1Rs. Future experiments will test this hypothesis.

Fatty acid binding proteins (FABPs) can bind and transport lipid molecules within and between cells (*Kaczocha et al., 2009*; *Ertunc et al., 2015*). One of these, adipocyte fatty-acid binding protein 4 (aP2), is secreted from adipocytes via EVs (*Ertunc et al., 2015*), and several FABPs are found in brain (*Owada et al., 1996*). Recent studies show that one of these proteins, FABP5, has high affinity for 2-AG, and its inhibition or genetic deletion impairs 2-AG-mediated signaling and plasticity at glutamate synapses in the dorsal raphe nucleus (*Haj-Dahmane et al., 2018*; *Owada et al., 1996*; *Kaczocha et al., 2012*). Based on these results, and our present observation that FABP5 is co-localized with the EV markers Iβ1 and flotillin-1 in EV fractions from the mouse midbrain, it is possible that 2-AG release may occur via binding to FABPs that are transported to the extracellular space via EVs, and therefore subject to mechanisms regulating EV secretion, such as Sig-1Rs, ARF6, and MLCK. Future studies will more closely examine this possibility to more completely understand the mechanisms of EV-dependent eCB release in the brain.

# Materials and methods

## Key resources table

| Reagent type | Designation | Source or reference | Identifiers | Additional information |
|---|---|---|---|---|
| Mouse: *M. musculus* (C57BL/6J) | C57BL/6J; wildtype, WT | Charles River Laboratories | Strain Code: 027 | |
| Mouse: *M. musculus* (C57BL/6J) | *sigma1r*; Sigma1 receptor: Sig-1R; Sig-1R KO, knockout | https://doi.org/10.1073/pnas.1518894112 | | |
| Mouse: *M. musculus* (C57BL/6J) | *Dagla fl/fl x Slc6a3-Cre +/-*; floxed DGL-α x DATCre heterozygote; DGL-α x DATCre; DGL-α KO, knockout | *Dagla fl/fl*, a gift from Sachin Patel; *Dagla fl/fl x Slc6a3-Cre + /-* breeders a gift from Daniel P. Covey | | |
| Mouse: *M. musculus* (C57BL/6J) | *CNR1*; CB1R; CB1R -/-; CB1R KO; knockout | https://doi.org/10.1073/pnas.96.10.5780 | | |
| Cell Line (*M. musculus*) | Mouse neuroblastoma x Rat glioma: NG108-15 cells; NG108 cells | ATCC | HB-12317 | |
| Antibody | Mouse monoclonal (mcl) anti-alpha-tubulin | Sigma-Aldrich | Cat#: T5168 | Western Blot (WB); Dilution (1:10,000) |
| Antibody | Rabbit polyclonal (plcl) anti-Alix | Sigma-Aldrich | Cat#: SAB4200476 | WB (1:1,000) |
| Antibody | Mouse monoclonal (mcl) anti-ARF6 | Santa Cruz Biotechnology | Cat#: sc-7971 | Immunohisto chemistry (IHC); (1:100), Immunopre cipitation (IP), 1 μg |
| Antibody | Rabbit plcl anti-ARF1 | Thermo Fisher Scientific | Cat#: PA1-127 | WB (1:1,000) |
| Antibody | Rabbit plcl anti-ARF6 | Cell Signaling Technology | Cat#: 3546 | WB (1:1,000) |
| Antibody | Mouse mcl anti-Cytochrome c | BD Biosciences | Cat#: 556433 | WB (1:1,000) |
| Antibody | Rabbit plcl anti-ERK1 | Santa Cruz Biotechnology | Cat#: sc-94 | WB (1:500) |
| Antibody | Rabbit mcl anti-FABP5 (D1A7T) | Cell Signaling Technology | Cat#: 39926 | WB (1:1,000) |
| Antibody | Rabbit plcl anti-Flotillin-1 | Santa Cruz Biotechnology | Cat#: sc-25506 | WB (1:1,000) |
| Antibody | Rabbit mcl anti-GAPDH (D16H11) | Cell Signaling Technology | Cat#: 5174 | WB (1:2000) |
| Antibody | Mouse mcl anti-GFP | Clonetech | Cat#: 632381 | WB (1:10,000) |
| Antibody | Rabbit plcl anti-GFP | Clonetech | Cat#: 632592 | IP (1 μg) |
| Antibody | Mouse mcl anti-Halo | Promega Corporation | Cat#: G9211 | WB (1:10,000) |

*Continued on next page*

Continued

| Reagent type | Designation | Source or reference | Identifiers | Additional information |
|---|---|---|---|---|
| Antibody | Mouse mcl anti-HSP90 | Enzo Life Sciences | Cat#: ADI-SPA-830 | WB (1:1,000) |
| Antibody | Mouse mcl anti-Integrin β1 | Thermo Fisher Scientific | Cat#: MA5-17103 | WB (1:1,000) |
| Antibody | Mouse mcl anti-Nucleoporin p62 | BD Biosciences | Cat#: 610498 | WB (1:1,000) |
| Antibody | Rabbit mcl anti-PDI | Cell Signaling Technology | Cat#: 3501 | WB (1:1,000) |
| Antibody | Rabbit anti-Sigma-1 receptor serum | A gift from Arnold Ruoho | N/A | IHC (1:1,000) |
| Antibody | Rabbit anti-Sigma-1 receptor serum #5460 | In house | N/A | WB (1:1,000) |
| Antibody | Mouse anti-sigma-1 receptor B-5 mcl | Santa Cruz Biotechnology | Cat#: Sc-137075 | IP (1 μg) |
| Antibody | Mouse mcl anti-Tyrosine hydroxylase | Millipore Corporation | Cat#: MAB318 | IHC (1:1,000), WB (1:2,000) |
| Antibody | Rabbit plcl anti-Tyrosine hydroxylase | Millipore Corporation | Cat#: AB152 | IHC (1:1,000) |
| Antibody | Chicken plcl anti-Tyrosine hydroxylase | Aves Labs | Cat#: TH | IHC (1:1,000) |
| Antibody | Mouse mcl anti-tsg 101 | Santa Cruz Biotechnology | Cat#: Sc-7964 | WB (1:500) |
| Antibody | Chicken plcl anti-vGAT | Synaptic Systems | Cat#: 131 006 | IHC (1:500) |
| Antibody | IRDye 680RD goat anti-mouse IgG | LI-COR Biosciences | Cat#: 925–68070 | WB (1:10,000) |
| Antibody | IRDye 800CW goat anti-mouse IgG | LI-COR Biosciences | Cat#: 925–32210 | WB (1:10,000) |
| Antibody | IRDye 680RD goat anti-rabbit IgG | LI-COR Biosciences | Cat#: 925–68071 | WB (1:10,000) |
| Antibody | IRDye 800CW goat anti-rabbit IgG | LI-COR Biosciences | Cat#: 925–32211 | WB (1:10,000) |
| Antibody | Alexa Fluor 405 goat anti-mouse IgG | Thermo Fisher Sci. | Cat#: A-31553 | IHC (1:500) |
| Antibody | Alexa Fluor 488 anti-chicken IgY | Thermo Fisher Sci. | Cat#: A-11039 | IHC (1:500) |

Continued

| Reagent type | Designation | Source or reference | Identifiers | Additional information |
|---|---|---|---|---|
| Antibody | Alexa Fluor 568 anti-rabbit IgG | Thermo Fisher Sci. | Cat#: A-11036 | IHC (1:500) |
| Recombinant DNA reagent | pcDNA3-CFP | A gift from Doug Golenbock | Addgene Plasmid # 13030 | |
| Recombinant DNA reagent | pARF6 (WT)-CFP | A gift from Joel Swanson; https://doi.org/10.1371/journal.pbio.0040162 | Addgene Plasmid # 11382 | |
| Recombinant DNA reagent | pARF6 (Q67L)-CFP | A gift from Joel Swanson; https://doi.org/10.1371/journal.pbio.0040162 | Addgene Plasmid # 11387 | |
| Recombinant DNA reagent | pARF6 (T27N)-CFP | A gift from Joel Swanson; https://doi.org/10.1371/journal.pbio.0040162 | Addgene Plasmid # 11386 | |
| Recombinant DNA reagent | pHTC HaloTag | Promega | Cat#: G7711 | |
| Recombinant DNA reagent | pHTN HaloTag | Promega | Cat#: G7721 | |
| Recombinant DNA reagent | Halo-Sig1R | This paper | N/A | contact for resource: Dr. Tsung-Ping Su; TSU@intra.nida.nih.gov |
| Recombinant DNA reagent | Sig1R-Halo | This paper | N/A | contact for resource: Dr. Tsung-Ping Su; TSU@intra.nida.nih.gov |
| Recombinant DNA reagent | Sig1R (1-60)-Halo | This paper | N/A | contact for resource: Dr. Tsung-Ping Su; TSU@intra.nida.nih.gov |
| Recombinant DNA reagent | Halo-Sig1R (61-223) | This paper | N/A | contact for resource: Dr. Tsung-Ping Su; TSU@intra.nida.nih.gov |
| Commercial assay or kit | NanoSight Particle Analysis | System Biosciences | Cat#: CSNANO100A-1 | |
| Commercial assay or kit | Dynabeads Protein G | Thermo Fisher Scientific | Cat#: 10009D | |
| Commercial assay or kit | PolyJet In Vitro DNA Transfection | Signagen Laboratories | Cat#: SL100688 | |
| Commercial assay or kit | Micro BCA Protein Assay Kit | Thermo Fisher Scientific | Cat#: 23235 | |
| Chemical compound, drug | Cocaine hydrochloride | NIDA Drug Supply | N/A | https://d14rmgtrwzf5a.cloudfront.net/sites/default/files/ndspcat24thedmarch2015.pdf |
| Chemical compound, drug | BD 1063 dihydrochloride | Tocris Bioscience | Cat#: 0883; CAS: 206996-13-6 | |
| Chemical compound, drug | SecinH3 | Tocris Bioscience | Cat#: 2849; CAS: 853625-60-2 | |
| Chemical compound, drug | AM251 | Tocris Bioscience | Cat#: 1117; CAS: 183232-66-8 | |

Continued

| Reagent type | Designation | Source or reference | Identifiers | Additional information |
|---|---|---|---|---|
| Chemical compound, drug | CGP55845 hydrochloride | Tocris Bioscience | Cat#: 1248; CAS: 149184-22-5 | |
| Chemical compound, drug | Hanks' Balanced Salt Solution | Thermo Fisher Scientific | Cat#: 14175095 | |
| Chemical compound, drug | Neurobasal Medium | Thermo Fisher Scientific | Cat#: 21103049 | |
| Chemical compound, drug | Collagenase | Thermo Fisher Scientific | Cat#: 17100017 | |
| Chemical compound, drug | Protease Inhibitor Cocktail | Sigma-Aldrich | Cat#: P8340 | |
| Chemical compound, drug | Blotting-grade blocker | Bio-Rad Laboratories | Cat#: 1706404 | |
| Chemical compound, drug | Bovine serum albumin | Sigma-Aldrich | Cat#: A2153 | |
| Chemical compound, drug | Percoll | GE Healthcare Life Sci. | Cat#: 17-0891-02 | |
| Chemical compound, drug | Dulbecco's Modified Eagle Medium | Thermo Fisher Scientific | Cat#: 11965092 | |
| Chemical compound, drug | Fetalgro Bovine Growth Serum | RMBIO | Cat#: FGR-BBT | |
| Chemical compound, drug | HAT Supplement (50X) | Thermo Fisher Scientific | Cat#: 21060017 | |
| Chemical compound, drug | Penicillin-Streptomycin (10,000 U/mL) | Thermo Fisher Scientific | Cat#: 15140122 | |
| Chemical compound, drug | Lauryl maltose neopentyl glycol | Anatrace | Cat#: NG310 | |
| Chemical compound, drug | two x Laemmli Sample Buffer | Bio-Rad Laboratories | Cat#: 1610737 | |
| Chemical compound, drug | Nonidet P-40 | Sigma-Aldrich | Cat#: I3021 | |
| Chemical compound, drug | Phenylme thanesulfonyl fluoride | Sigma-Aldrich | Cat#: P7626 | |
| Chemical compound, drug | NAV2729 | Tocris Bioscience | Cat#: 5986; CAS: 419547-11-8 | |
| Chemical compound, drug | ML seven hydrochloride | Tocris Bioscience | Cat#: 4310; CAS: 110448-33-4 | |
| Chemical compound, drug | NE100 | Tocris Bioscience | Cat#: 3313; CAS: 149409-57-4 | |

*Continued on next page*

*Continued*

| Reagent type | Designation | Source or reference | Identifiers | Additional information |
|---|---|---|---|---|
| **Software, algorithm** | | | | |
| GraphPad Prism 7 | | GraphPad Software, San Diego, CA | | |
| Image Studio Lite | | L LI-COR Biosciences, Lincoln, Nebraska | | |
| WINLTP 2.30 | | WinLTP Ltd., Bristol, U.K. | | https://www.winltp.com/ |
| G-Power 3.1.9.4 | | https://doi.org/10.3758/BF03193146 | | http://www.psychologie.hhu.de/arbeitsgruppen/allgemeine-psychologie-und-arbeitspsychologie/gpower.html |

## Drugs

1-[2-(3,4-Dichlorophenyl)ethyl]−4-methylpiperazine dihydrochloride (BD 1063 dihydrochloride, Cat#: 0883, Tocris), and cocaine hydrochloride were dissolved in 0.9% NaCl. N-[4-[5-(1,3-Benzodioxol-5-yl)−3-methoxy-1H-1,2,4-triazol-1-yl]phenyl]−2-(phenylthio)acetamide (SecinH3, Cat#: 2849, Tocris) was dissolved in DMSO, and then diluted with 25% DMSO/75% glucose solution (5 w/v%).

## Animals

### Ethics statement

All animal procedures were conducted in accordance with the principles as indicated by the *NIH Guide for the Care and Use of Laboratory Animals*. These animal protocols were also reviewed and approved by the NIDA intramural research program Animal Care and Use Committee, which is fully accredited by the Assessment and Accreditation of Laboratory Animal Care (AAALAC) International (approved protocols: 17-CNRB-15, 16-CNRB-128, 16-INB-1, 16-INB-3, 17-INB-5).

Adult (8+ weeks) male mice were housed with food and water available ad libitum. Mice were housed on a 12/12 hr light cycle. Wild-type C57Bl6/J mice were ordered from Charles River Laboratories. Sigma one receptor transgenic mice were bred in house. *Sigmar1* mutant (+/−) *Sigmar1*[Gt(IRE-SBetageo)33Lex] litters on a C57BL/6J × 129s/SvEv mixed background were purchased from the Mutant Mouse Regional Resource Center at the University of California, Davis. The sigma-1 receptor (+/−) males were backcrossed for 10 generations to female on C57BL/6J to ensure that animals had a homogenous background. The resulting mice were genotyped to select Sig-1R WT and KO mice. To generate mice lacking diacylglycerol lipase-α (DGL-α) in DA neurons, mice in which the *Dagla* gene was flanked by *LoxP* were obtained from the laboratory of Dr. Sachin Patel (Vanderbilt University). These mice were then crossed with dopamine transporter (*Slc6a3*; DAT) *Cre* mice (*Slc6a3*[Cre+/-]) to generate mice lacking the DGL-α gene (*Dagla*) in DAT-expressing neurons (*Dagla*[fl/fl] x *Slc6a3-Cre*[+/-]).

## Group allocation

Group membership was determined by genotype where transgenic mice were used. In in vitro electrophysiology studies, recordings from untreated control brain slices were interleaved with recordings from drug pre-incubated brain slices from the same animal. In cell biology experiments, mice were chosen for experiments depending upon date of arrival from the supplier. In this way, mice were assigned to groups according availability and to the experimental procedures to be performed that day. In most cases, brain tissue from each mouse was used in both control and treatment conditions. NG-108 cell culture dishes were selected randomly from those available in the tissue incubator.

## Isolation of mouse midbrain slices

Mice were killed with $CO_2$ gas, and brains were removed, and rinsed in ice-cold Hank's balanced salt solution (Thermo Fisher Scientific). Midbrain samples were isolated by cutting coronal sections containing the VTA using mouse brain matrices (Roboz), and the cortex and a hippocampus dissected free (*Figure 2—figure supplement 1*).

## Preparation of EV fractions

### NG108 cells

To isolate EVs from NG108 cells we used an established protocol with minor modifications (*Gabrielli et al., 2015a*). First, conditioned HBSS was collected and pre-cleared from cells and debris by centrifugation at 300 x g for 10 min, and 2000 x g for 10 min. Then, for EV purification, the supernatant was centrifuged at 100,000 x g for 60 min. Pellets obtained from this spin-down were then resuspended in 30 µL of lysis buffer (50 mM Tris, pH 7.4, 150 mM NaCl, 1% Triton-X and protease inhibitor (Sigma-Aldrich) for western blotting. Cocaine stimulation occurred by adding the drug (1–10 µM) to the cultures in HBSS.

### Midbrain

For vesicle fractions from brain tissue we used an established protocol with minor modifications (*Perez-Gonzalez et al., 2012*; *Polanco et al., 2016*). Briefly, following dissection, midbrain slices from two wildtype male C57BL/6J mice were chopped and then incubated in 1.5 ml of 0.125% collagenase (Sigma-Aldrich) in Neurobasal medium (Thermo Fisher Scientific) for 30 min at 37°C (see *Figure 2—figure supplement 1* for a graphic summary of Ev isolation procedures). To stop the digestion, 4.5 ml of ice-cold phosphate-buffered saline (PBS) was added and the temperature maintained at 4°C throughout subsequent steps. The tissue was then gently disrupted by multiple passes through a 200 µL pipette tip, followed by a series of differential centrifugations at 300 x g for 10 min, 2000 x g for 10 min, and 7500 x g for 30 min. The pellets resulting from these spins, containing cells, membranes, and cellular debris, respectively, were then discarded. For EV purification, the 7500 x g supernatant was syringe filtered at 1.0 µm (Whatman Puradisc Syringe Filters, GE Healthcare Life Sciences, Cat. #6780–2510) and centrifuged at 100,000 x g for 70 min to obtain a pellet containing EVs. The 100,000 x g pellet was washed with PBS and spun again at 100,000 x g for 60 min to obtain a total EV (tEV) pellet. For EV purification, the tEV sample was resuspended in 0.5 mL of 0.95 M sucrose in 20 mM HEPES (pH 7.4) before addition to a sucrose-step gradient column. The column consisted of 6 × 0.5 mL fraction running from the bottom 2.0 M, 1.65 M, 1.3 M, 0.95M, 0.6 M, to 0.25 M at the top. Similarly, sucrose step gradients were centrifuged for 16 hr at 200,000 x g, after which the six fractions were collected. EVs settled typically at 0.95 M sucrose. The original six 0.5 mL fractions were collected and resuspended in 6 mL of ice-cold PBS, followed by a 100,000 x g centrifugation for 70 min at 4°C. Finally, the pellets were resuspended in 30 µL of filtrated-PBS when EVs were used for cell assays or 15 µl of lysis buffer (50 mM Tris pH7.4, 150 mM NaCl, 1% Triton-X and protease inhibitor (Sigma-Aldrich) when EVs were intended for western blots. For western blotting, EV lysates in lysis buffer were quantified for protein content with a Micro BCA Protein Assay Kit (Thermo Fisher Scientific). We also prepared brain lysate sample (BL) in lysate buffer using the midbrain tissues from the 300 x g pellets obtained in the courses of the EV isolations, which were used as positive controls for the western blots and to normalize tEVs sample amount between each treatment.

## Drug treatment regimen

Drugs were injected i.p. at a volume of 5 ml/kg. Regimen 1 (for *Figure 2E*): Thirty and 60 min after i.p. injections with cocaine (15 mg/kg), midbrain slices were collected. Regimen 2 (for *Figures 2F* and *4E*): Injections with BD1063 (10 mg/kg, s.c.), SecinH3 (10 µmol/kg, s.c.), ML7 (5 mg/kg), or vehicle (inj 1) were performed 20 min prior to injections with saline or cocaine (15 mg/kg, i.p.; inj 2). Thirty min after inj 2, midbrain slices were collected. Regimen 3 (for *Figure 4A*): 10, 20 and 30 min after i.p. injections with cocaine (15 mg/kg), midbrain slices were collected. Regimen 4 (for *Figure 4B*): Injections with BD1063 (10 mg/kg, s.c.), SecinH3 (10 µmol/kg, s.c.), or vehicle (s.c.) (inj 1) were performed 20 min prior to injections with saline or cocaine (15 mg/kg, i.p.; inj 2). Ten min after inj 2, midbrain slices were collected. Regimen 5 (for *Figure 4C*): 20 min after i.p. injections with

cocaine (15 mg/kg) or vehicle, midbrain slices were collected. Regimen 6 (for *Figure 5B*): 30 min after i.p. injections with cocaine (15 mg/kg) or vehicle, midbrain slices were collected. For western blotting of extracellular vesicles from NG108 cells, the cells on 10 cm dishes were washed with pre-warmed Hanks' Balanced Salt Solution (HBSS) twice and incubated in HBSS at 37℃ in the presence of cocaine.

## Western blotting

In brief, western blotting was performed with protein samples separated using a 12% sodium dodecyl sulfate-polyacrylamide gel electrophoresis (SDS-PAGE), and then transferred onto a Immo-bilon FL Transfer polyvinylidene difluoride (PVDF) membrane (Mollipore) in the Tris/Glycine buffer (Bio-Rad Laboratories) without methanol. After incubation with 5% blotting-grade blocker (Bio-Rad Laboratories) or 5% bovine serum albumin (BSA, Sigma-Aldrich) in TBST buffer (10 mM Tris. pH 8.0, 150 mM NaCl, and 0.5% Tween 20) for 1 hr, membranes were incubated with the primary antibodies at 4℃ overnight. Membranes were washed for 10 min four times by using TBST buffer and incubated with a 1:10,000 dilution of secondly antibodies (LI-COR Biosciences) at room temperature for 1 hr. Blots were washed for 10 min four times by using TBST buffer and the signal intensity was deter-mined using Odyssey Imaging System (LI-COR Biosciences). Resultants were analyzed using an Image Studio Lite (LI-COR Biosciences).

## Nanoparticle tracking analysis (NTAs) for EVs

Total EV (tEV) samples were isolated in filtered (at 1 μm)-PBS from WT and Sig1R KO mouse mid-brain, 30 min after treatment with either saline or cocaine (15 mg/kg, i.p.), and sent to Systems Bio-sciences (Palo Alto, CA) for metric analysis of tEVs.

## Isolation of MAM from mouse midbrain tissues

MAM was isolated from mouse mid brain as previously reported (*Hayashi and Su, 2007*; *Kourrich et al., 2013*). Briefly, following homogenization of the brain tissue, nuclear, crude mito-chondrial, and microsomal fractions were prepared by differential centrifugation. Supernatants were collected as the cytosolic fraction. The crude mitochondrial fraction in the isolation buffer (250 mM mannitol, 5 mM HEPES, 0.5 mM EGTA, pH 7.4) was subjected to a Percoll gradient centrifugation for separation of the MAM from mitochondria.

## Immunofluorescence staining

Immunofluorescence staining was performed as described previously. In brief, after blocking, the sections were incubated with the first antibodies in 5% BSA/0.1% Triton X-100 PBS overnight at 4℃. Bound antibodies were detected with Alexa Fluor 405-conjugated anti-mouse IgG (1:200, Thermo Fisher Scientific), Alexa Fluor 488-conjugated anti-chicken IgG (1:200, Thermo Fisher Scientific), and Alexa Fluor 568-conjugated anti-Rabbit IgG antibodies (1:200, Thermo Fisher Scientific) in 5% BSA PBS. An UltraView confocal microscopic system (PerkinElmer) was used for imaging.

For the immunostaining of Sig-1R, rabbit anti-serum against Sig-1R, a gift from Dr. Arnold Ruoho (University of Wisconsin, USA; *Ramachandran et al., 2007*), was used. When compared to several commercially available products, the affinity-purified antibody from this antiserum, is very specific for the sigma-1 receptor in the mouse dorsal root ganglia (*Mavlyutov et al., 2016*). We established the following procedures to allow for the best specific detection of the Sig-1R in mouse brain slices, using the antiserum from Dr. Ruoho. Deeply anesthetized animals were transcardially perfused with filtered 0.1 M Phosphate buffer (PB; pH 7.4) followed by 4% paraformaldehyde (w/v) in 0.1 M PB. After perfusion, whole brains were isolated and post-fixed in the same fixatives overnight at 4℃ with rotation. Subsequently, they were dehydrated with 20% sucrose in 0.1 M PB (w/v) and then 30% sucrose in 0.1 M PB (w/v) at 4℃ with rotation. The brain samples were then embedded in O.C.T. compound (Sakura Finetek, Torrance, CA) on dry ice and stored in −80℃. Thirty-μm sections were cut on a cryostat and mounted on Tissue Path Superfrost Plus Gold Microscope Slides (Fisher Scien-tific, Hamilton, NH) dried overnight. Sections were blocked with 5% bovine serum albumin (BSA, w/v) in PBS containing 0.1% Triton-X100 (v/v) for 1 hr at room temperature. The sections were then incubated with the sigma-1 receptor anti-sera diluted at 1:1000 in the blocking solution overnight at 4℃. Following 10 min PBS washing for three times, sections were incubated with Alexa Fluor (488

for green/568 or 594 or 546 for Red)-conjugated goat anti-rabbit IgG (1:500, Invitrogen, Carlsbad, CA) in 5% BSA in PBS for 90 min at room temperature. The sections were washed with PBS for 5 min three times, then counterstained with 4',6'-diamino-2-phenylindole (DAPI, Invitrogen, 1 μg/mL in MilliQ; Millipore, Billerica, MA) by 10 min incubation at room temperature. Sections were washed with PBS for 5 min three times, mounted on coverslips with Prolong Diamond Antifade Mountant (Life technologies, Carlsbad, CA) for imaging. The specificity of this antiserum in labeling the Sig-1R is demonstrated in brain slices from wildtype mice, where strong staining is shown, and in and Sig-1R knockout mice, where staining is absent (*Figure 3—figure supplement 1*).

## Immunoprecipitation

### Brain tissue

The midbrain slice sample was homogenized in 900 μl of ice-cold IP lysis buffer-1 (50 mM Tris pH7.4, 150 mM NaCl, 0.1% lauryl maltose neopentyl glycol (Anatrace, Maumee, OH) and protease inhibitors (Sigma-Aldrich) with a glass Dounce homogenizer (20 strokes). After centrifugation at 15,000 g for 10 min, protein concentration of cellular extracts was measured using a Micro BCA Protein Assay Kit (Thermo Fisher Scientific). Five hundred μg of protein amount in supernatants were mixed with ice-cold IP lysis buffer-1 with protease inhibitors to adjust total 1000 μl. The samples were incubated and rotated with 5 μg ARF6 (Santa cruz) antibody at 4°C for overnight. Forty μl of prewashed Dynabeads Protein G (Thermo Fisher Scientific) added into the sample, incubated and rotated at 4°C for 90 min. Immunoprecipitants were washed five times with 0.8 ml of ice-cold IP lysis buffer-1 for 5 min. Samples were boiled in 30 μl elution buffer, which is combined between 15 μl of 2 x Laemmli Sample Buffer (Bio-Rad Laboratories) and 15 μl 7 M Urea/1% CHAPS at 37°C for 10 min. Importantly, 2-mercaptoethanol was omitted from the endogenous Sig1R IP assay to prevent degrading antibody disulfide bonds. Proteins were analyzed with a 12% SDS-PAGE.

### NG-108 cells

All processes were performed on ice. The overexpressed NG108 cells in 100 mm dishes were washed twice with cold PBS and then lysed in 1.0 ml of IP lysis buffer-2 (50 mM Tris pH7.4, 150 mM NaCl, 1% Nonidet P-40 (Sigma-Aldrich) and protease inhibitors (Sigma-Aldrich). After centrifugation at 15,000 g for 10 min, protein concentration of cellular extracts was measured using a Micro BCA Protein Assay Kit (Thermo Fisher Scientific). One-hundred fifty μg of supernatants were mixed with PBS in equal volume. The supernatants were incubated and rotated at 4°C overnight with 1 μg of the rabbit anti-EGFP/EYFP/ECFP (Clontech) or 1 μg normal rabbit IgG (Santa Cruz). Thirty ml of prewashed Dynabeads Protein G (Thermo Fisher Scientific) was then applied, and samples were rotated for 90 min at 4°C. Immunoprecipitants were washed 4 times with 0.8 ml of IP lysis buffer-2 for 5 min, and twice with 1 ml of PBS for 5 min. Samples were boiled in 70 μl elution buffer combined between 35 μl of 2 x Laemmli Sample Buffer with 5% 2-ME and 35 μl lysis buffer at 95°C for 5 min. Proteins were analyzed with a 12% SDS-PAGE.

## Cell culture and transfection

NG108 cells were cultured at 37°C and 5% $CO_2$ in High glucose Dulbecco's Modified Eagle Medium (DMEM, Thermo Fisher Scientific) containing L-glutamine, 10% Fetalgro Bovine Growth Serum (RMBIO), HAT supplement (Thermo Fisher Scientific), 100 mg/ml Penicillin-Streptomycin (Thermo Fisher Scientific). Transfection of cells with expression vectors was done by using PolyJet DNA In Vitro Transfection Reagent (Signagen Laboratories, Rockville, MD) according to manufacturer's instructions. Sources of vectors are provided above.

## Measurement of 2-AG in brain tissue

### 2-AG extraction

2-Arachidonoyl glycerol (2-AG) was extracted from samples using a modified Folch extraction method. A mixture of chloroform/methanol (2:1 v/v) was added to the sample at a rate of 8 μL for each μg of protein detected. An internal standard 10Z-heptadecenoylethanolamide (HEA, 17:1 ethanolamide, Avanti Polar Lipids, Alabaster, Al) (4 μg/mL) was included in this volume and was added at a rate of 0.05 μL per μg of protein. Samples were homogenized, sonicated and vortexed. Two μL of water was added for each μg of protein in the sample. The mixture was again vortexed and

centrifuged. The extraction results in an upper aqueous phase and a lower organic phase (containing 2-AG and the internal standard, HEA 17:1 ethanolamide). The lower phase (organic phase) was evaporated to dryness using nitrogen, re-suspended in 500 µL of chloroform and fractionated. The procedure used for fractionation was similar to one developed previously for eCBs (*Schmid et al., 2000*). The fractionation was performed with Discovery SPE-Si tubes 1 mL (Sigma-Aldrich, St. Louis, MO). The samples were loaded on the columns in 500 µL chloroform and then washed with 3 mL of chloroform. Next, the 2-AG was eluted with 3 mL of chloroform/methanol (98/2%). Finally, the elute was evaporated to dryness using nitrogen and re-suspended in 100 µL of acetonitrile.

## Mass spectrometry analysis

Samples were diluted 1:1 (v/v) in 400 µm silver acetate in acetonitrile prior to mass analysis. A previous study has demonstrated the advantages to adding silver cations into the sample mixture for detecting 2-AG (*Kingsley and Marnett, 2003*). Samples were analyzed on an Oribtrap Velos (Thermo Fisher) in positive ion mode with a static nanospray source with 4 µm spray tips and a capillary temperature of 200°C. The Fourier transform mass spectrometry (FTMS) mode with a mass resolution of 100K was employed for all samples. The mass error for 2-AG assignment was ±3 ppm and MS/MS analyses were also conducted to confirm the identification of 2-AG.

## In vitro electrophysiology

Twelve-week-old WT C57BL6, *Cnr1*$^{-/-}$ (CB1R knockout), or *Sigmar1*$^{+/-}$ Sig-1R KO mice were decapitated, and their brains rapidly removed and transferred to an oxygenated (95% $O_2$, 5% $CO_2$) ice-cold solution containing (in mM) 93 N-Methyl-D-glucamine (NMDG), 2.5 KCl, 1.2 $NaH_2PO_4$, 30 $NaHCO_3$, 20 HEPES, 25 Glucose, 3 Sodium pyruvate, 10 $MgCl_2$, 0.5 $CaCl_2$, 5.6 Ascorbic acid. Horizontal slices (220 µm) containing the VTA were sectioned using a Leica VT1200S vibratome (Leica Biosystems) and transferred to a holding chamber at room temperature (RT) filled with oxygenated solution containing (in mM) 109 NaCl, 4.5 KCl, 1.2 $NaH_2PO_4$, 35 $NaHCO_3$, 20 HEPES, 11 Glucose, 1 $MgCl_2$, 2.5 $CaCl_2$, 0.4 Ascorbic acid. After incubation for at least 1 hr in the holding chamber at RT, slices were transferred to a recording chamber perfused with oxygenated aCSF containing (in mM) 126 NaCl, 3 KCl, 1.2 $NaH_2PO_4$, 26 $NaHCO_3$, 11 Glucose, 1.5 $MgCl_2$, 2.4 $CaCl_2$, maintained at 35–36° C using an inline solution heater (Warner Instruments, Hamden, CT). Cells were visualized with an upright microscope (Olympus BX51WI) equipped with infrared interference-contrast optics. Recorded neurons identified in the lateral VTA, medial to the terminal nucleus of the accessory optic track (MT) and anterior to the third cranial nerve. Dopamine neurons were identified in the lateral VTA using electrophysiological criteria in cell-attached mode. Only cell demonstrating regular pacemaker firing (>3 Hz) and action potential widths > 2.5 ms were chosen for further recording (*Ungless and Grace, 2012*). Whole-cell voltage-clamp recordings from DA neurons were acquired using an Axopatch 200B amplifier (Molecular Devices, San Jose, CA). Recording pipettes (3.5–5 MΩ) were pulled with a P-97 horizontal micropipette puller (Sutter Instruments, Novato, CA) and filled with internal solution containing (in mM) 140 K-gluconate, 2 NaCl, 1.5 $MgCl_2$, 10 HEPES, 10 Tris-phosphocreatine, 4 Mg-ATP, 0.3 Na-GTP, 0.1 EGTA (pH 7.2, 290 mOSM). DNQX (20 µM), DL-AP5 (40 µM), picrotoxin (100 µM) and strychnine (1 µM) were present in the aCSF to block AMPA, NMDA, $GABA_A$ and glycine receptors, respectively. Electrophysiological identification of DA neurons was performed in cell-attached mode to select only cells exhibiting pacemaker firing and action potential widths < $GABA_B$ IPSCs were evoked using electrical stimulation with bipolar tungsten stimulating electrodes with tip separation of 300–400 µm. A train of 6 stimuli of 100µs duration were delivered at 50 Hz every 30 s. Stimulation protocols were generated, and signals acquired using the electrophysiology software WinLTP. Control $GABA_B$ currents were recorded for 10 min before the appropriate drug was applied for an additional 30 min. Data was analyzed using WinWCP software (Courtesy of Dr. John Dempster, Strathclyde University, Glasgow, UK). Figures were generated, and statistics analyzed using GraphPad Prism6 (v6.07; LaJolla, CA). Data are presented as the change in percent from control.

## Quantification, statistical analysis and reporting

The experiments were designed using estimates of effect size and standard error derived from prior experience and pilot experiments. These values were then used in power analysis calculations using

the program G-Power (version 3.1.9.4, University of Dusseldorf, Germany) to determine sample sizes. Means ± s.e.m. are used throughout to report measures of centricity and dispersion. A spreadsheet (*Source data 1*) describing means, significance levels and 95% confidence intervals for each experiment is included with this report. Statistical tests were determined by the number of groups and treatments to be compared. An omnibus test was used when necessary statistical assumptions could be met. Thus, in experiments where repeated measures could be obtained from the same subjects, samples, or cells (e.g. time course data), a repeated-measures ANOVA was used. When repeated measures were not performed, and group size was >2, a one-way ANOVA was used. Posthoc analyses (Tukey's, Dunnett's, or Bonferroni's multiple comparison tests) were determined by the type of omnibus test, as well as the nature of the multiple comparisons (pairwise rows and columns, comparison to control columns, main effects versus interactions). When only two groups of data were compared, a Student's t-test was used. In all cases, a two-tailed p value of 0.05 was considered the minimum for significance. Actual p values are reported for all omnibus tests, unless $p < 0.0001$, and the statistical information is reported in the figure captions. In immunoprecipitation experiments, co-localization was determined from observed association on Western blots, and therefore, statistical tests were not used (*Figure 1F and G*; *Figure 3B and D*; *Figure 5D*; *Figure 1—figure supplement 1C*).

## Acknowledgements

This work is supported by the US Department of Health and Human Services, National Institutes of Health, and National Institute on Drug Abuse, Intramural Research Program. YN was supported in part by the Japanese Society for Promotion of Sciences. We would like to acknowledge expert assistance of Dr. Shiliang Zhang of the NIDA-IRP Confocal and Electron Microscopy Core, as well as the efforts of Dr. Francois Vautier, Director, of the NIDA-IRP Breeding Facility, and all of NIDA-IRP the transgenic facility staff.

## Additional information

### Funding

| Funder | Grant reference number | Author |
| --- | --- | --- |
| National Institute on Drug Abuse | 1ZIADA000487-14 | Carl R Lupica |
| National Institute on Drug Abuse | 1ZIADA000206-33 | Tsung-Ping Su |
| Japan Society for the Promotion of Science | | Yoki Nakamura |

The funders had no role in study design, data collection and interpretation, or the decision to submit the work for publication.

### Author contributions

Yoki Nakamura, Conceptualization, Data curation, Formal analysis, Validation, Investigation, Methodology, Writing—original draft, Writing—review and editing; Dilyan I Dryanovski, Conceptualization, Validation, Investigation, Methodology, Project administration, Writing—review and editing; Yuriko Kimura, Data curation, Formal analysis, Investigation; Shelley N Jackson, Resources, Formal analysis, Validation, Investigation, Visualization, Methodology; Amina S Woods, Conceptualization, Resources, Data curation, Formal analysis, Supervision, Investigation, Methodology, Project administration; Yuko Yasui, Data curation, Validation, Investigation, Methodology; Shang-Yi Tsai, Formal analysis, Investigation, Visualization, Methodology; Sachin Patel, Resources, Funding acquisition, Validation, Writing—review and editing; Daniel P Covey, Resources, Data curation, Validation, Methodology, Writing—review and editing; Tsung-Ping Su, Conceptualization, Resources, Data curation, Formal analysis, Supervision, Funding acquisition, Investigation, Methodology, Project administration, Writing—review and editing; Carl R Lupica, Conceptualization, Resources, Data curation, Formal analysis, Supervision, Funding acquisition,

Validation, Investigation, Methodology, Writing—original draft, Project administration, Writing—review and editing

## Author ORCIDs
Sachin Patel http://orcid.org/0000-0001-8052-520X
Daniel P Covey http://orcid.org/0000-0001-9596-108X
Carl R Lupica https://orcid.org/0000-0002-5375-3263

## Ethics

Animal experimentation: Ethics Statement: All animal procedures were conducted in accordance with the principles as indicated by the NIH Guide for the Care and Use of Laboratory Animals. These animal protocols were also reviewed and approved by the NIDA intramural research program Animal Care and Use Committee, which is fully accredited by the Assessment and Accreditation of Laboratory Animal Care (AAALAC) International (approved protocols: 17-CNRB-15, 16-CNRB-128, 16-INB-1, 16-INB-3, 17-INB-5).

## Decision letter and Author response
Decision letter https://doi.org/10.7554/eLife.47209.021
Author response https://doi.org/10.7554/eLife.47209.022

## Additional files

### Supplementary files
• Source data 1. Statistical Data.
DOI: https://doi.org/10.7554/eLife.47209.018
• Transparent reporting form DOI: https://doi.org/10.7554/eLife.47209.019

### Data availability
All data generated or analysed during this study are included in the manuscript and supporting files.

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
