## [Decision Letter]

Thank you for submitting your article "Cocaine-induced endocannabinoid signaling mediated by sigma-1 receptors and extracellular vesicle secretion" for consideration by *eLife*. Your article has been reviewed by three peer reviewers, and the evaluation has been overseen by a Reviewing Editor and Gary Westbrook as the Senior Editor. The following individuals involved in review of your submission have agreed to reveal their identity: Ken Mackie (Reviewer #1); Christopher Ford (Reviewer #2); Nephi Stella (Reviewer #3). The reviewers have discussed the reviews with one another and the Reviewing Editor has drafted this decision to help you prepare a revised submission.

Summary:

This study examines the hypothesis that cocaine stimulates 2-AG secretion through its interactions with the sigma-1 receptor (Sig-1R) leading to release of EVs (enriched) in 2-AG, which then engage CB1 receptors.

Essential revisions:

The revised version of this study must fully address the main concerns of reviewer #1. I want to emphasize that this is indispensable and understand that additional experiments may need to be carried on. Thus, experimental evidence showing that EVs are secreted by cocaine stimulated NG108-15 and that the cocaine stimulated 2-AG increase is specific for EVs and not general for all midbrain membranes must be provided (see verbatims of the reviewer's comments below). The editors and reviewers during their discussion were in full agreement on these essential revisions.

Reviewer #1:

In this interesting and provocative study, Nakamura and colleagues examine the hypothesis that cocaine stimulates 2-AG secretion through its interactions with the sigma-1 receptor (Sig-1R) leading to release of EVs (enriched) in 2-AG, which then engage CB1 receptors.

1) Central to their hypothesis is that cocaine causes release of EVs containing 2-AG. To demonstrate release, it is necessary to isolate them from the media of the NG108-15 cells (as was done in the Gabrielli study). From a careful reading of the Materials and methods, it does not seem that this was done. Rather, NG108-15 cells were washed, scraped, homogenized and membranes with EV markers enriched. Thus, what is established by the NG108-15 studies is that the various signature of EVs are enriched within NG108-15 cells by cocaine, but not that the EVs are secreted.

2) Indirect support for 2-AG being secreted by EVs is that 2-AG concentration, when normalized to membrane protein content, is increased in EVs compared to 2-AG present in total midbrain membranes. It is preferable to normalize lipids to total lipid content or mole% of lipids in the preparation to control for different protein content in EVs vs. total membrane. In support of the central hypothesis, it is important to demonstrate that the increase in 2-AG following cocaine is specific for EVs and not generally for all midbrain membranes.

3) Figure 5 (in combination from earlier work by this group) nicely shows that GABA IPSC inhibition likely results from 2-AG produced by DA neuronal DAGL-α. Is the rather large increase in midbrain 2-AG all due to increase in 2-AG from DA neurons? This should be evaluated by measuring 2-AG in midbrain in the DAGL-α conditional KOs after cocaine treatment.

4) That inhibition or genetic deletion of Sig-1R slows, but does not block cocaine inhibition of IPSCs is quite interesting. The authors establish that CB1 responses are not altered by Sig-1R manipulations, but do Sig-1R manipulations affect IPSC inhibition when 2-AG is synthesized by other routes? For example, will the combination of DHPG/phenylephrine still produce 2-AG to reduce IPSCs after inhibition of Sig-1Rs? Establishing whether Sig-1Rs are needed for all synaptic release of 2-AG would greatly increase the impact of the findings. Similarly, does this synapse show DSI? If so, will manipulations of Sig-1R prevent DSI? (The phenomenon shown here is reminiscent of inhibition of DSI, but not LTD.)

5) While the authors interpret the inhibition of the rapid, but not the delayed, phase of cocaine IPSC inhibition as potential support for a "readily releasable pool" of 2-AG, an alternative interpretation is that manipulating Sig-1R affects DAGL localization to the synapse. This should be examined in the Sig-1R KO.

Reviewer #2:

In the present manuscript by Nakamura et al., the authors examine a novel pathway by which cocaine regulates the release of 2-AG. Over a series of experiments, the authors outline a pathway by which cocaine activates the Sig-1R, which releases a tonic inhibition on ARF6. This small G-protein when in the GTP bound state activates MLCK which in turn increases non-synaptic extracellular vesicle release. In the first part of the manuscript the authors utilize NG-108 cells as well as midbrain homogenates to elucidate this pathway. They find that cocaine can increase markers of EVs suggesting that cocaine increases EV release in a Sig-1R dependent manner. Utilizing Western blots, co-IPs and immunostaining they then go on to show that in the midbrain cocaine decreases the association of Sig1R and ARF6 and that the GEF inhibitor SH3 inhibits cocaine-induced EV release. Finally to examine the functional consequence of cocaine on EV release they utilize slice electrophysiology. The authors find that the mechanism by which cocaine leads to the increase in 2AG release (as measured by presynaptic inhibition of GABAB IPSCs) may partially involve this pathway as the effects were blocked by sigma-1 receptor antagonists as well as ARF6 and MLCK blockers.

The experiments are thorough, have been done well and interpreted appropriately. Furthermore, the combination of approaches come together nicely to clearly define this new mechanism by which cocaine may regulate endocannabinoid signaling. Based on volume of work presented I have no further suggestions for other experiments as the results and conclusions are clear. A potential caveat is that the ARF6 and MLCK blockers could have additional effects, not dependent on EV release, which could also be driving 2AG release. However further confirming this pathway and true EV release would likely involve a significant amount of further work, beyond the scope of this manuscript.

A minor point, however is that in many of the figures, summary data are only presented for some experiments. It is unclear at least from the results and figure legends if the other experiments are an n = 1 or if they are only representative blots from experiments that have been replicated several times. A second point is that in Figure 4D, the authors show that cocaine increases GTP-bound ARF6 in the midbrain. Presumably this relies on a specific antibody that only recognizes GTP-ARF6 however it was not clear that this antibody can distinguish between GDP and GTP bound forms.

Reviewer #3:

This study demonstrates the involvement of a novel cellular compartment (non-synaptic extracellular vesicle (EVs) and its intracellular machinery (Sig-1R > ARF6) in 2-AG-mediated activation of CB1R. The experimental design is thorough and uses validated pharmacological and genetic tools in established model systems. The technical approaches are sound, the manuscript clearly written, and conclusions and interpretations based on convincing results. I believe that this study will have an important impact on the field of cannabinoid research and neuroscience in general, and provide an important foundation for future studies.

1) Figure 1—figure supplement 1A and B. The authors should provide the fold change in expression of Sig-1R to help the reader better understand the impact of this result.

2) Figure 2—figure supplement 1: Mention in the text the time between stimulation with cocaine and EV harvesting to help the reader understand this result.

3) Figure 3A and 3C: the magnified images are not convincing and should be improved. Their scale bars are missing, and it appears that the magnification is merely 2X. Higher quality and magnification images should be provided here.

4) Figure 3E: the diagram could be increased in size and its clarity improved (e.g. reduce size/thickness of the X that overlays ARF6).

5) Figure 4D: The qualities of the images should be greatly improved, and the authors should provide magnifications that clearly illustrate TH versus ARF6-GTP locations.

[Editors' note: further revisions were requested prior to acceptance, as described below.]

Thank you for resubmitting your work entitled "Cocaine-induced endocannabinoid signaling mediated by sigma-1 receptors and extracellular vesicle secretion" for further consideration at *eLife*. Your revised article has been favorably evaluated by Gary Westbrook as the Senior and Reviewing Editor, and three reviewers. The manuscript has been improved but there a few remaining issues that we would like addressed before acceptance, as outlined below: We expect to handle the revised manuscript at the editorial level.

Summary:

The reviewers were generally satisfied with the revisions of your manuscript and agreed that it provides a significant conceptual advance in understanding the mechanisms by which cocaine regulates 2AG release via the sigma-1 receptor. The authors clarified that EVs were purified from NG108-15 culture medium. They were also able to demonstrate the requirement of Sig-1R in phenylephrine/DHPG 2-AG-mediated IPSC inhibition, while there appears to be no role for Sig-1R in "basal" 2-AG-mediated CB1 activation. However the reviewers had a few additional comments that should be addressed in a final revision.

1) The authors were not able to demonstrate an increase in midbrain EV 2-AG levels following cocaine administration (possibly due to the sensitivity of their 2-AG measurement technique coupled with the small number of EVs that can be purified from midbrain), which undermines a major premise of the study-cocaine stimulates release of 2-AG enriched EVs to inhibit IPSCs. This limitation needs to be appropriately integrated into their Results and Discussion. Perhaps an alternative explanation that should be considered is that there is not more 2-AG per EV, but there are more EVs produced, following cocaine or phenylephrine/DHPG and this is sufficient to activate CB1 to inhibit GABA release.

2) Emphasis in the Discussion. a) There needs to be a significant shift in the emphasis of the paper's Discussion. Because Sig-1R is required for phenylephrine/DHPG 2-AG-mediated IPSC inhibition (Figure 7), the emphasis of cocaine binding to Sig-1R as a causal mechanism in mid-brain eCB IPSC inhibition appears misplaced. If anything, in the cocaine treated slices, Sig-1R is only necessary for the short term (<20 minutes) cocaine/eCB-mediated depression of IPSCs (Figure 6D) as IPSC inhibition by cocaine seems similar in the wildtype and Sig-1R KOs by 30 minutes (Figure 6D). In contrast (Figure 7B), for phenylephrine/DHPG 2-AG-mediated IPSC inhibition, Sig-1R KO reduces IPSC inhibition over the entire 30 minutes of drug application. An alternative to the mechanism proposed in the Discussion (cocaine binds to the Sig-1R…, and Figure 8—figure supplement 1) is that cocaine increases NE, activating α_1_Rs, which then require Sig-1R, ARF6, etc., to release EVs (or stimulate DAGLA activity). The experimental data in midbrain slices that cocaine needs to bind to sig1-Rs for IPSC inhibition does not seem conclusive, only that Sig-1R and ARF6 are required at some point in the pathway.

b) Similarly, the difference in the role of Sig-1R, ARF6, MLCK between cocaine and phenylephrine/DHPG inhibition of IPSCs needs to be briefly discussed. As mentioned in the Discussion, these proteins seem to play a role in early IPSC inhibition by cocaine, but not late inhibition. On the other hand, at least Sig-1Rs are required for both early and late IPSC inhibition by phenylephrine/DHPG. Can the authors integrate these findings into their model?

3) Figure 5C: The WB results showing FLOT-1 in EV has a "dark dot" in its lane which limits the interpretation of the result. The authors should provide another representative result.

---

## [Author Response]

Reviewer #1:[…] 1) Central to their hypothesis is that cocaine causes release of EVs containing 2-AG. To demonstrate release, it is necessary to isolate them from the media of the NG108-15 cells (as was done in the Gabrielli study). From a careful reading of the Materials and methods, it does not seem that this was done. Rather, NG108-15 cells were washed, scraped, homogenized and membranes with EV markers enriched. Thus, what is established by the NG108-15 studies is that the various signature of EVs are enriched within NG108-15 cells by cocaine, but not that the EVs are secreted.

We appreciate the reviewer mentioning this oversight on our part. We neglected to state in the original Materials and methods that we followed the protocol published in the Gabrielli et al. paper, with only a few modifications. Thus, we collected HBSS from the cultures after treatment with cocaine (and other manipulations) and then subjected this effluent to EV isolation procedures (sequential centrifugation and sucrose gradients to yield fr3). Therefore, the EV markers were isolated from fractions collected from the cells after stimulation, and not from homogenized cells. We now include a more complete description of these methods in the manuscript (subsection “Midbrain”).

2) Indirect support for 2-AG being secreted by EVs is that 2-AG concentration, when normalized to membrane protein content, is increased in EVs compared to 2-AG present in total midbrain membranes. It is preferable to normalize lipids to total lipid content or mole% of lipids in the preparation to control for different protein content in EVs vs. total membrane. In support of the central hypothesis, it is important to demonstrate that the increase in 2-AG following cocaine is specific for EVs and not generally for all midbrain membranes.

Total lipids were not measured in the mass spectrometry assay as we only measured 2-AG and our internal standard. This is because it is not possible to measure total lipid content using MS. It is possible to record a larger m/z range which permits detection of a broader range of lipid species, but this would still only be a small fraction of the total lipid content. Also, it is worth noting here that the methods in the Gabrielli et al. paper indicate that anandamide levels were also normalized to total protein content.

We also agree that measuring cocaine-induced changes in 2-AG in EV fractions using FTMS would provide specific and direct evidence of our hypothesis. However, after conducting this experiment several times we have encountered several limitations. First and foremost, the amount of EV material obtained in fr3 from ventral midbrain and available for analysis using FTMS is extremely small, necessitating the pooling of fr3 samples from several mice. To achieve this, we must first kill the mice 15 min after injection of saline or cocaine (3 mice per sample), then rapidly dissect the ventral midbrain from each mouse and pool samples across mice to quickly perform centrifugation and sucrose gradient procedures. We have now performed these experiments on three separate occasions in three different cohorts of mice (n = 30 mice), and we observe a high degree of variability in both baseline levels of 2-AG using FTMS, and in the level of the cocaine-induced increase in 2-AG levels. In general, there is a trend toward increased 2-AG in fr3 in the cocaine-injected mice (see Author response image 1), but this does not reach statistical significance because of the high degree of variability across cohorts. It is likely that some of this variability results from the difficulty of the dissection procedure (identifying and dissecting only ventral midbrain from a whole a mouse brain) and to other factors that we have not yet identified. Therefore, at present we are confident that we can identify 2-AG in the EV-containing fr3 from mouse midbrain, but we do not have the statistical power to demonstrate a significant increase in 2-AG content of these fractions after cocaine stimulation. We defer to the discretion of the reviewer and the editors as to whether these data should be included in the manuscript.

3) Figure 5 (in combination from earlier work by this group) nicely shows that GABA IPSC inhibition likely results from 2-AG produced by DA neuronal DAGL-α. Is the rather large increase in midbrain 2-AG all due to increase in 2-AG from DA neurons? This should be evaluated by measuring 2-AG in midbrain in the DAGL-α conditional KOs after cocaine treatment.

A previous study, that we now cite in the manuscript, used in-situ hybridization, immunohistochemistry and electron microscopy to localize DGLα in most neurons in the VTA, including DA and non-DA neurons (Mátyás et al., 2008, Identification of the sites of 2-arachidonoylglycerol synthesis and action imply retrograde endocannabinoid signaling at both GABAergic and glutamatergic synapses in the ventral tegmental area. Neuropharmacology, 54:95-107). Therefore, it is likely that 2-AG is synthesized in several cell types in the VTA. However, we reason that because we measure the inhibition of IPSCs in only identified DA neurons, 2-AG release from these other sources may not be detected at synapses onto these cells because of the limited spatial range over which 2-AG can act before its uptake or metabolism occurs. Alternatively, it is possible that the non-DA neurons that express DGLα do not release 2-AG when exposed to cocaine because they lack α_1_-noradrenergic receptors, or some other necessary component.

Although these are undoubtedly interesting issues, the observation that the 2-AG effect on synaptic transmission is absent in the DGLα KOs, as well as in the brain slices treated with the DGLα inhibitor tetrahydrolipostatin (Figure 5—figure supplement 1), strongly suggest that the endocannabinoid involved in the inhibition is 2-AG. Since the identification of the endocannabinoid was the primary goal of these experiments, we feel that this has been achieved. Moreover, these transgenic mice are not yet freely available, and were included in this study under a material transfer agreement restricting their use to only DA neurons.

4) That inhibition or genetic deletion of Sig-1R slows, but does not block cocaine inhibition of IPSCs is quite interesting. The authors establish that CB1 responses are not altered by Sig-1R manipulations, but do Sig-1R manipulations affect IPSC inhibition when 2-AG is synthesized by other routes? For example, will the combination of DHPG/phenylephrine still produce 2-AG to reduce IPSCs after inhibition of Sig-1Rs? Establishing whether Sig-1Rs are needed for all synaptic release of 2-AG would greatly increase the impact of the findings. Similarly, does this synapse show DSI? If so, will manipulations of Sig-1R prevent DSI? (The phenomenon shown here is reminiscent of inhibition of DSI, but not LTD.)

These are excellent suggestions, and we have conducted additional experiments to address some of these issues. First, as indicated by the reviewer, our prior work in rat VTA shows that the increase in 2-AG function produced by cocaine could be replicated by combined application of DHPG and phenylephrine (PE) to activate type-I metabotropic glutamate receptors (mGluRI) and α_1_-noradrenergic receptors, respectively. Therefore, if this effect can be replicated in mice, we would predict that this effect of DHPG+PE should, like that of cocaine, be inhibited when Sig-1R function is impaired or absent, and this would provide a strong case that Sig-1Rs and the EV signaling cascade is important for GPCR-dependent 2-AG release in the VTA. In these new experiments we found that the inhibition of IPSCs produced by DHPG+PE was very similar to that produced by cocaine alone, and that the effect of DHPG+PE was significantly reduced in DA neurons from Sig-1R knockout mice. Moreover, the effects of DHPG+PE were blocked in mouse DA neurons, as reported in rats. Therefore, this supports the involvement of Sig-1Rs in GPCR-dependent 2-AG release. We have now included these data in a new figure (Figure 7A-C) and discuss them in the paper (subsection “Sig-1R antagonism prevents cocaine-stimulated 2-AG function in VTA DA neurons” and Discussion, last paragraph).

Regarding the second part of the reviewer’s question, we have only established that DSI occurs at GABA inputs to VTA DA neurons in THCre mutant rats, using cre-driven channel rhodopsin to initiate DSI in the DA neuron (see Wenzel et al., Curr Biol., 28:1392-1404, 2018). However, we have not performed these experiments in mice because of limitations in obtaining THCre mice. However, we have also reported that tonic inhibition of IPSCs by 2-AG is observed in both rat and muse DA neurons, and this is uncovered when CB1Rs are blocked by AM251, or when DGLα is blocked by THL (Riegel and Lupica, 2004; Wang et al., 2015). Although we have not verified that this source of 2-AG is 100% independent of GPCR stimulation, it is present under baseline conditions in the absence of cocaine, or other exogenously applied GPCR agonists. Therefore, in new experiments we measured this tonic 2-AG response in the mouse VTA and observed the expected robust increase in IPSCs during AM251 application. Importantly, this level of tonic 2-AG inhibition did not differ between wildtype and Sig-1R knockout mice. This, together with the DHPG+PE data, suggests that Sig-1Rs and EVs are primarily involved in promoting 2-AG release derived from GPCR stimulation-dependent synthesis. We have compiled these data obtained with the Sig-1R KO and WT mice and have added them to a new Figure 7D-E, describe them the Results section, and expanded the Discussion a bit to deal with the distinction in Sig-1R modulation of GPCR-dependent 2-AG function.

5) While the authors interpret the inhibition of the rapid, but not the delayed, phase of cocaine IPSC inhibition as potential support for a "readily releasable pool" of 2-AG, an alternative interpretation is that manipulating Sig-1R affects DAGL localization to the synapse. This should be examined in the Sig-1R KO.

Although this is an interesting possibility, it would probably best be addressed using electron microscopy to examine the localization of DGLα in the VTA of Sig-1R KO mice. Alternatively, evidence that DGLα function is intact in the Sig-1R KOs would suggest that the enzyme is not altered in these mutant mice. Prior work from our lab shows that 2-AG is tonically present at GABAergic synapses on DA neurons in the rat VTA, and its influence on IPSCS can be observed when CB1Rs are antagonized by AM215, resulting in an increase in the IPSC response (Riegel and Lupica, 2004; Wang et al., 2015, cited in the manuscript). If this tonic 2-AG release was not altered in the Sig-1R KO mouse VTA, then this would provide support for intact DGLα function. Therefore, we have performed additional experiments comparing the effect of AM251 on IPSCs in wildtype and Sig-1R KO mouse VTA and find no difference. Therefore, it appears that functionally, with respect to tonic 2-AG, there is no change in DGLα function in the mutant mice. *We have added an additional figure (Figure 7) to the manuscript showing the impaired DHPG+PE-induced 2-AG effect (comment #4), and intact tonic 2-AG function in the Sig-1R KO mice. We have also added this information to the Results and Discussion sections.*

Reviewer #2:[…] The experiments are thorough, have been done well and interpreted appropriately. Furthermore, the combination of approaches come together nicely to clearly define this new mechanism by which cocaine may regulate endocannabinoid signaling. Based on volume of work presented I have no further suggestions for other experiments as the results and conclusions are clear. A potential caveat is that the ARF6 and MLCK blockers could have additional effects, not dependent on EV release, which could also be driving 2AG release. However further confirming this pathway and true EV release would likely involve a significant amount of further work, beyond the scope of this manuscript.A minor point, however is that in many of the figures, summary data are only presented for some experiments. It is unclear at least from the results and figure legends if the other experiments are an n = 1 or if they are only representative blots from experiments that have been replicated several times.

This information is included in the metadata file that has been revised to add newly-included experiments. We have also now added this information to the relevant figure legends.

A second point is that in Figure 4D, the authors show that cocaine increases GTP-bound ARF6 in the midbrain. Presumably this relies on a specific antibody that only recognizes GTP-ARF6 however it was not clear that this antibody can distinguish between GDP and GTP bound forms.

The product description for the antibody (New East Biosciences, Malvern, PA, catalog number 26918) indicates that it can distinguish ARF6-GTP from ARF6-GDP. However, as we are unable to use this antibody for quantitative Western blot analyses of ARF6-GTP, and we have confirmed co-localization of ARF6 with TH and the Sig-1R in the mouse midbrain in Figure 3, we have elected to remove this image (Figure 4D) and descriptions of measurement of ARF6-GTP from the manuscript.

Reviewer #3:[…] 1) Figure 1—figure supplement 1A and B. The authors should provide the fold change in expression of Sig-1R to help the reader better understand the impact of this result.

We agree with the reviewer that this information is important to include, and we have added it to the Figure 1—figure supplement 1A caption. The siRNA significantly reduced Sig-1R expression by approximately 50% (49 ± 12% of control, p < 0.0001, unpaired t-test). However, a description of the overexpression of the Halo-tagged Sig-1R in Figure 1—figure supplement 1B is somewhat meaningless because this receptor is not natively expressed (see “Halo-”, 3rd row Western blot Figure 1—figure supplement 1B). Therefore, the x-fold change will be deceptively high and is not reported.

2) Figure 2—figure supplement 1: Mention in the text the time between stimulation with cocaine and EV harvesting to help the reader understand this result.

These times vary according to the experiment and are always mentioned in the figure legends and text describing these experiments.

3) Figure 3A and 3C: the magnified images are not convincing and should be improved. Their scale bars are missing, and it appears that the magnification is merely 2X. Higher quality and magnification images should be provided here.

We have performed additional immunohistochemical experiments and have provided new images at higher resolution. The scale bar is now 20 µm, rather than 50 µm as in the original figure.

4) Figure 3E: the diagram could be increased in size and its clarity improved (e.g. reduce size/thickness of the X that overlays ARF6).

We have re-worked this schematic to improve clarity.

5) Figure 4D: The qualities of the images should be greatly improved, and the authors should provide magnifications that clearly illustrate TH versus ARF6-GTP locations.

See response to reviewer 3. We have removed this image from the manuscript.

[Editors' note: further revisions were requested prior to acceptance, as described below.]1) The authors were not able to demonstrate an increase in midbrain EV 2-AG levels following cocaine administration (possibly due to the sensitivity of their 2-AG measurement technique coupled with the small number of EVs that can be purified from midbrain), which undermines a major premise of the study-cocaine stimulates release of 2-AG enriched EVs to inhibit IPSCs. This limitation needs to be appropriately integrated into their Results and Discussion. Perhaps an alternative explanation that should be considered is that there is not more 2-AG per EV, but there are more EVs produced, following cocaine or phenylephrine/DHPG and this is sufficient to activate CB1 to inhibit GABA release.

We have expanded the Results section to include these data and have added them to a revised Figure 5 (Figure 5C). In addition, we now consider this information in the Discussion and have added the alternative explanation provided by the reviewer.

2) Emphasis in the Discussion. a) There needs to be a significant shift in the emphasis of the paper's Discussion. Because Sig-1R is required for phenylephrine/DHPG 2-AG-mediated IPSC inhibition (Figure 7), the emphasis of cocaine binding to Sig-1R as a causal mechanism in mid-brain eCB IPSC inhibition appears misplaced. If anything, in the cocaine treated slices, Sig-1R is only necessary for the short term (<20 minutes) cocaine/eCB-mediated depression of IPSCs (Figure 6D) as IPSC inhibition by cocaine seems similar in the wildtype and Sig-1R KOs by 30 minutes (Figure 6D). In contrast (Figure 7B), for phenylephrine/DHPG 2-AG-mediated IPSC inhibition, Sig-1R KO reduces IPSC inhibition over the entire 30 minutes of drug application. An alternative to the mechanism proposed in the Discussion (cocaine binds to the Sig-1R…, and Figure 8—figure supplement 1) is that cocaine increases NE, activating α_1_Rs, which then require Sig-1R, ARF6, etc., to release EVs (or stimulate DAGLA activity). The experimental data in midbrain slices that cocaine needs to bind to sig1-Rs for IPSC inhibition does not seem conclusive, only that Sig-1R and ARF6 are required at some point in the pathway.

In re-considering these data, we noted differences between the inhibition of IPSCs produced by DHPG+PE and cocaine that we feel might explain this and we have included this in the manuscript in the form of new information in the Results and Discussion, and a new Figure 7—figure supplement 1. Specifically, we noted that the kinetics of the response to DHPG+PE differed from that of cocaine, and we reason that the faster onset and offset of the cocaine effect could reflect direct activation of Sig-1Rs that initiates rapid EV release and depletions. On the other hand, we suggest that the slower kinetics of the DHPG+PE effect could reflect delayed recruitment of EVs by an intracellular messenger release by GPCR activation, and we cite a recent study showing that choline can act as a Sig-1R agonist to perhaps fulfill this role.

b) Similarly, the difference in the role of Sig-1R, ARF6, MLCK between cocaine and phenylephrine/DHPG inhibition of IPSCs needs to be briefly discussed. As mentioned in the Discussion, these proteins seem to play a role in early IPSC inhibition by cocaine, but not late inhibition. On the other hand, at least Sig-1Rs are required for both early and late IPSC inhibition by phenylephrine/DHPG. Can the authors integrate these findings into their model?

See response to part “a” above.

3) Figure 5C: The WB results showing FLOT-1 in EV has a "dark dot" in its lane which limits the interpretation of the result. The authors should provide another representative result.

This has been remedied by providing another WB from another similar experiment.